# Acidification impacts and acclimation potential of Caribbean benthic foraminifera assemblages in naturally discharging low-pH water

Daniel François[1,a,*], Adina Paytan[2], Olga Maria Oliveira de Araújo[3], Ricardo Tadeu Lopes[3], Cátia Fernandes Barbosa[1].

[1]*Departamento de Geoquímica, Universidade Federal Fluminense, Niterói, Brazil.*
[2]*Institute of Marine Sciences – University of California, Santa Cruz, USA.*
[3]*Federal University of Rio de Janeiro (UFRJ), Nuclear Instrumentation Laboratory, Nuclear Engineering Program/COPPE, Rio de Janeiro, Brazil*
[a]*now at: Department of Ocean Systems, NIOZ Royal Netherlands Institute for Sea Research and Utrecht University, Texel, the Netherlands*

*Correspondence author: Daniel François (danielfrancois@id.uff.br)

**Abstract**. Ocean acidification (OA) is expected to negatively affect many ecologically important organisms. Here we report the response of Caribbean benthic foraminiferal assemblages to naturally discharging low-pH waters with a composition similar to that expected for the end of the 21st century. At low-pH ~7.8 and low saturation state with respect to calcite ($\Omega_{calcite} < 4$), the relative abundance of hyaline, agglutinated and symbiont-bearing species increased, indicating higher resistance to potential carbonate chemistry changes. Diversity and other taxonomical metrics (i.e., richness, abundance and evenness) declined steeply with decreasing pH despite exposure of this ecosystem to low pH conditions for millennia, suggesting that tropical foraminiferal communities will be negatively impacted under acidification scenarios SSP3-7.0 and SSP5-8.5. The species *Archaias angulatus*, a major contributor to sediment production in the Caribbean was able to calcify at more extreme conditions (7.1 pH) than those projected for the late 21st century, but the calcified tests had a lower average density than those exposed to higher pH conditions (7.96), indicating that reef foraminiferal carbonate production might decrease this century. Smaller foraminifera were particularly sensitive to low pH and our results demonstrate their potential use to monitor OA conditions.

## 1 Introduction

With anthropogenic carbon dioxide ($CO_2$) emissions steadily increasing since the beginning of the industrial age, levels are currently higher than they have been in the past 800,000 years (Petit et al., 1999; Lüthi et al., 2008). Global emissions are increasing annually and lead to a continued uptake of $CO_2$ by the oceans. Consequently, surface ocean pH and saturation state decreases ($- 0.0181 \pm 0.0001$ decade$^{-1}$, Lida et al., 2021), a process commonly known as ocean acidification (OA) (Doney et al., 2020). Based on the Coupled Model Intercomparison Project Phase Six (CMIP6), a further decrease of surface ocean pH is expected for all Shared Socioeconomic Pathways (SSPs) at the end of the 21st century (Kwiatkowski et al., 2020; IPCC, 2021). Because the carbonate system has major control on biogenic calcification efficiency this process is expected to negatively affect many ecologically important calcifying organisms such as corals (Kroeker et al., 2013; Crook et al., 2013; Hughes et al., 2017), foraminifers (Uthicke, Momigliano, and Fabricius, 2013; Kawahata et al., 2019), and coralline crustose algae (Penã et al., 2021).

Among these, foraminifera are dominant members of both planktonic and benthic communities with widespread distribution in the ocean. They are vital to calcium carbonate ($CaCO_3$) cycling (Langer et al., 1997; Langer, 2008) and on a global scale, they are estimated to contribute a total of 14 billion tons of $CaCO_3$ per year, accounting for approximately 25 % of the current total $CaCO_3$ production (Langer, 2008).

Due to their ability to consume substantial amounts of organic matter, they are also relevant for organic carbon cycling (Moodley et al., 2000), and they constitute a key link in marine food webs. Once they died, their shells (also known as tests) become important contributors to sediment accumulation in many ecosystems (Yamano, Miyajima, and Koike, 2000; Doo et al., 2016) and hence relevant for the carbon burial flux in the ocean (Schiebel, 2002). With ongoing OA and future scenarios projecting further changes

(Kwiatkowski et al., 2020; IPCC, 2021), it is vital for assessing biological feedbacks and changes in biochemical cycles to understand how foraminifera will be affected. Many studies under controlled conditions documented the negative impact of lower pH on calcification, weight, size, and taxonomical metrics (Nehrke et al., 2013; Kawahata et al., 2019; Narayan et al., 2021, and references therein). However, some studies have also demonstrated either resilience (Engel et al., 2015; Pettit et al., 2015; Stuhr et al.,

2021), or even positive effects on foraminifera, such as enhanced calcification (Fujita et al., 2011) or enzymatic calcification activity (Prazeres et al., 2015), demonstrating the complexity of foraminiferal responses to OA. Additionally, relatively little is known about how foraminifera respond in natural settings with low-pH low carbonate saturation conditions, which is crucial for determining if and how communities have the potential to acclimate.

In situ investigations have been performed in natural $CO_2$ vents in the Mediterranean Sea (Dias et al., 2010; Pettit et al., 2015), Papua New Guinea (Uthicke, Momigliano, and Fabricius, 2013), the northern Gulf of California (Pettit et al., 2013) and coastal springs in Puerto Morelos (PM), Mexico (Martinez et al., 2018). In the latter, recruitment and early succession (Crook et al., 2016), acclimatization potential (Crook et al., 2013), and the responses of calcifying communities were studied (Crook et al., 2012; Martinez et al., 2018),

demonstrating that despite general deleterious effects, some organisms were able to calcify under OA conditions. A study focused on large benthic foraminifera (LBF) reported that porcelaneous, chlorophyte-bearing foraminifera (e.g., *Archaias angulatus*) were relatively less impacted (Martinez et al., 2018). Study sites such as coastal springs allow the investigation of foraminiferal communities under projected future conditions more realistically, helping to decrease the uncertainty in global-scale models. However, a

detailed survey considering community-wide responses (i.e., including smaller foraminifera) is necessary to ascertain a wider range of potential impacts.

As $CO_2$ emissions continue to grow despite emerging climate policies (Peters et al., 2020), global awareness has demonstrated a strong interest in research focused on potential impacts for mitigative action. To build on and expand the findings at PM we aimed to (i) explore the mid-term (i.e., multidecadal) responses of

foraminifera species to OA using total assemblages, (ii) investigate the effects of OA on assemblages of both large and small foraminifera for acidification scenarios projected to the end of the 21st century (Kwiatkowski et al., 2020; IPCC, 2021), (iii) explore the taphonomical and ecological implications of *post mortem* alterations for reef ecosystems, and (iv) investigate possible acclimation patterns in the shell structure of the species *A. angulatus*. Specifically, an examination of assemblage structure, taxonomic

metrics, assemblage test size, preservation potential and an X-ray micro-CT analysis in the species *A. angulatus* were employed.

## 2 Methods

### 2.1 Study site and data retrieval

The Yucatán Peninsula is a karstic region in Southern Mexico (Fig. 1a) where Tertiary limestones are
underlain by an ejecta/evaporite complex. Several structural and tectonic features divide the area into six distinct physiographic regions (Back and Hanshaw, 1970). Among these, Puerto Morelos reef lagoon is part of the Holbox Fracture Zone–Xel-Ha region, which is characterized by >100 km long chain of elongated depressions referred to as 'sabanas' (Perry, Velazquez-Oliman, and Marin, 2002). In this area, rainwater infiltrates the porous karstic limestone (Fig. 1b) and flows towards the ocean through
interconnected caves and fractures where the groundwater mixes with seawater in the underground aquifers before discharging between the shore and the offshore barrier reef (Beddows et al., 2007; Null et al., 2014). Flowing through the limestone and interacting with the strata through processes of dissolution, precipitation, and mixing, the groundwater conditions change and the resulting low-pH, low carbonate saturation state ($\Omega$), and high inorganic C content water discharges along the Mexican Caribbean coast
(Back and Hanshaw, 1970; Perry, Velazquez-Oliman, and Marin, 2002; Crook et al., 2012, 2013, 2016; Martinez et al., 2018, 2019; Hernandez-Terrones et al., 2021). The submarine groundwater discharges at submarine springs, which structure ranges from long ''fractures'' to small circular depressions "seeps" (Fig. 1d, spring Agua), (Crook et al., 2012). The discharge of the springs is relatively constant throughout the year (Crook et al., 2016), and the lagoon circulation is not significantly affected by tides (av. 17 cm),
and currents due to the microtidal regime of the region (Coronado et al., 2007) and the springs location in the protected back-reef. The waves overtopping the reef is the main driving factor of circulation, which is generally slow (av. 2–3 cm s$^{-1}$), with faster (av. 20 cm s$^{-1}$) flow restricted to the northern and southern channels where the water exits the lagoon (Coronado et al., 2007). At the springs, the discharged waters with slightly lower salinity mainly flows vertically and not towards the sediment due to the buoyancy effect.
The beach sediments in the area are composed of coarse (~0.258 mm) carbonate sands of biogenic origin (Escudero et al. 2020).

Surface sediment samples (< 1 cm depth) were retrieved using a plastic spoon at various distances from the center of six submarine springs (Fig. 1c, Gorgos, Laja, Mini, Norte, Agua, and Pargos) in October 2011. In the laboratory, samples were stained (rose Bengal, 1 g/L ethanol), weighed, washed with deionized water
through a 63 µm sieve mesh, and dried at 50 °C for 24 hours. Discrete water samples were also retrieved near the sites of sediment collection for chemical analysis. Carbonate chemistry, temperature, and salinity data from seven samples reported in Martinez et al.(2018) were complemented with 20 additional samples collected at the same day following the protocols described by the authors. Briefly, the samples were filtered (0.2 µm) and split into aliquots for the analysis of salinity, total inorganic carbon ($C_T$) and total alkalinity
($A_T$), following the protocols of Dickson, Sabine and Christian (2007). The $C_T$ was measured on a CM5011 Carbon Coulometer (UIC, Inc.; analytical measurement error: $\pm$ 3 µmol kg$^{-1}$) and $T_A$ using an automated open-cell, potentiometric titrator (Orion model 950; analytical measurement error: $\pm$ 2 µmol kg$^{-1}$). Salinity

was measured using a portable salinometer (Portasal Model 8410, Guild Line). Seawater temperature was measured in situ with a handheld YSI micro-processor (Yellow Springs model 63). The pH (seawater scale), carbonate ion concentration ($CO_3^{2-}$) and calcite saturation state ($\Omega_{calcite}$) were calculated using the program CO$_2$Sys (Pierrot, Levis and Wallace, 2006), using the CO$_2$ dissociation constants of Lueker, Dickson and Keeling (2000); KHSO$_4$ – Dickson, Sabine and Christian (2007); and B concentration – Uppström, 1974. Certified CO$_2$ reference material (from A. Dickson lab at UC San Diego, batch 112) was used to calibrate all instruments.

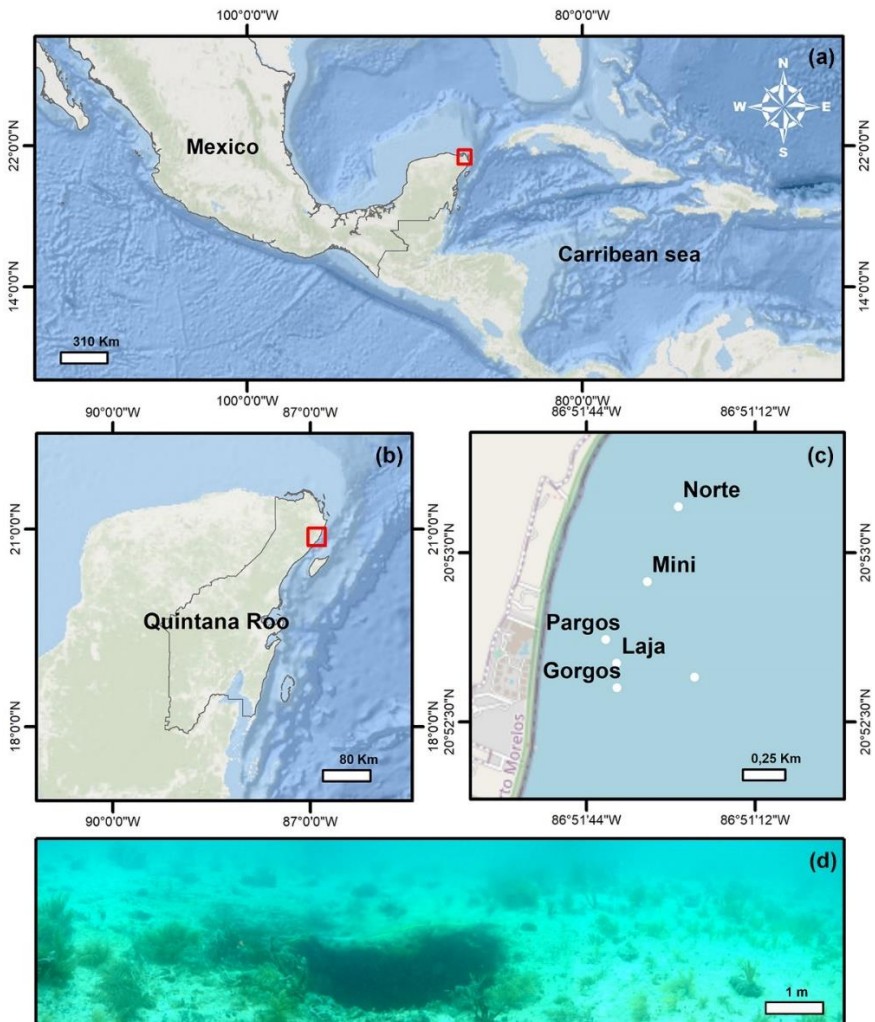

**Figure 1** (a) Location map of the Yucatán Peninsula, (b) Quintana Roo, and (c) the six submarine springs (Gorgos, Laja, Mini, Norte, Pargos and Agua) studied at Puerto Morelos reef Lagoon (National Marine Park), (d) Spring agua, which structure presents a small circular depression.

**2.2 Foraminiferal analysis**

Sediment dry weight was recorded, and samples were homogenized and split into several small aliquots to allow efficient picking. The specimens found in each pre-weighted sediment aliquot were counted under a

Zeiss STEMI 2000 stereomicroscope until a minimum of 250 specimens were recorded for each sample. Foraminiferal tests were identified to the lowest possible taxonomic level and assigned to informal species categories for diversity analyses. The taxonomic classification was based on the specialized bibliography of Cushman (1929), Jones (1994), and supplementary taxonomic studies (Milker and Schmiedl, 2012; Abu-Zied, Al-Dubai, and Bantan, 2016; Sariaslan and Langer, 2021). Each species and genus were verified against the World Register of Marine Species (WoRMS, 2022) to ensure the use of the most recent nomenclature.

The samples were stained in rose Bengal to consider the living counts, while most tests were at least partially stained the proportions of fully stained specimens were small (~3 %) hence total (live plus dead) assemblages were used. We expect that the sample represents accumulation over several decades. This approach allows us to assess the mid-term responses of foraminiferal assemblages since the generational accumulation of tests in sediments integrate the effects of stressors over time (Hallock et al., 2003). This smooths out any seasonal fluctuations and therefore allows to use the foraminifera responses to average environmental conditions (Scott and Medioli, 1980). We note that the low live percentage is a common pattern as most reef-dwelling taxa tend to live on phytal or hard substrates rather than directly on the sediments (Martin, 1986; Barbosa et al., 2009, 2012; Stephenson, Hallock and Kelmo, 2015). Shannon-Weiner Diversity Index (H′), and Pielou's evenness (J′) were calculated considering the standardized foraminifera density for a volume of 1 cm$^3$ sediment. These taxonomic metrics were calculated as follows: Shannon-Weiner Diversity Index with the equation $H′=-\Sigma(Pi*log(Pi))$, where Pi is the proportion of individuals per species; Pielou's evenness with the equation $J′=H′/log(S)$, where H′ is the Shannon-Weiner Diversity Index and S the species richness. Assemblage distributions were assessed according to differences in functional groups, i.e., symbiont-bearing and opportunistic, and test type groups, i.e., small miliolids, small rotaliids, and agglutinated that do not present an opportunistic behavior. This approach has been used by Amergian et al. (2022) in nearby settings, based on categories designed by Hallock et al. (2003) for sensitivity/stress-tolerance taxa and by Murray (2006) for different test compositions.

## 2.3 Taphonomy and assemblage test size analysis

To analyze alteration by taphonomy, foraminiferal tests were classified into three categories 'optimally' (i.e., pristine tests), 'well' (i.e., tests with weak taphonomic signals), and 'poorly' (i.e., strongly abraded or fragmented tests) preserved, following the descriptions of Yordanova and Hohenegger (2002). Discoloration patterns were analyzed to detect any vertical mixing and exposure of relict tests. In general, if colored black (with iron/manganese sulfides) the tests indicate relict sediments deposited under reducing conditions, whereas a brown coloration indicates the oxygenation of iron through the reworking of the sediments (Maiklem, 1967); white tests indicate lack of significant sediment burial and alteration. For a complete survey of the assemblage test size distribution, the surface area of all individuals was calculated using the ImageJ software (Schneider, Rasband, and Eliceiri, 2012). All specimens picked were placed on the dorsal side in common brass picking trays and photographed under the same magnification and camera settings using an adapter for a microscope camera (Prazeres et al. 2015), to trace surface area changes (i.e., gain or loss) in large benthic foraminiferal species under low-pH conditions. Specifically, the test area was

defined according to the gray scale differences between the surface of the individuals test (white) and background (black). The surface area parameter was the most suitable for the analysis considering the high taxonomical and consequently morphological diversity of PM samples since it identifies the size of the foraminifera tests in a standard way.

**2.4 X-ray MicroCT**

An X-ray MicroCT analysis was performed on four individuals from ambient (7.96) and from low-pH conditions (7.11). To ensure that the analyzed tests represent living conditions, only tests in excellent condition, and therefore, not influenced by *post mortem* processes of dissolution and transport were selected. For the X-ray microCT acquisition, a V/TOMEX/M (GE Measurement and Control Solutions, Wunstorf, Germany) was used. The microCT parameters for the acquisition included a voltage of 60 kV,

current of 100 μA, 5 frames, and an Al filter with a thickness of 0.5 mm. The geometry had a magnification of 31.81 and pixel size of 6.28 μm. Certified calcite standards were used to calibrate the density for the analyzed samples. The 3D reconstructions were performed using the Phoenix Datos X Reconstruction software, in which a slice alignment, beam hardening correction was implemented, and a mathematical edge-enhancement filter was applied to achieve a higher contrast at the edges of the chambers. For the 3D

visualization, VG Studio Max v 3.0 and Avizo 2020.3 software packages were used. For calcite density analysis, the CTAnalyser v. 1.18.4.0 software was used. Calcite density was assessed by the calcite density distribution calculated from the CT number that was determined based on the X-ray attenuation coefficient of each sample (Iwasaki et al., 2019). In addition, estimation of morphometric parameters such as total volume and chamber wall thickness distribution were performed.

**2.5 Statistical analysis**

A BIO-ENV procedure (9999 permutations) and global BEST test (statistical significance) were used to identify the set of explanatory environmental parameters that produced a Euclidean matrix that best correlated (Spearman method) the species assemblage similarity matrix and normalized environmental variables. Polynomial models (second order) were performed to investigate the relationships between

carbonate chemistry and the taxonomical metrics (n = 26). They were compared according to their contribution to the model's Akaike Information Criterion (AIC), and the models with the lowest AIC value (i.e., highest fit) were selected for the analysis. For comparison of *A. angulatus* microstructure parameters between high and low-pH the student's t-test (n = 8) was used for variables with normal distributions and homogenous variances. When these conditions were not met, Welch's t-test was performed. We used the

Kruskal-Wallis test to assess differences between functional groups, taxonomic metrics, and assemblage test size. For the latter, the stations were separated into four pH groups: 8.1−8.05 pH representing present-day conditions (n = 4); 8.01−7.9 pH aligned with low-intermediate acidification scenarios SSP1-2.6 and SSP2-4.5 (n = 11); 7.8−7.7 representing high acidification scenarios SSP3-7.0 and SSP5-8.5 (n = 4); 7.6−7.2 representing acidification conditions beyond those predicted for the end of $21^{st}$ century (n = 7).

Data normality and variance homogeneity were tested using Shapiro-Wilk and Levene's Test. The BIO-ENV and global BEST procedures were performed in Primer v.6 software (Clarke and Gorley, 2006).

Student's t-test, Welch's t-test, Kruskal-Wallis test, and data visualization were performed using R software (version 4.0.2; http://www.Rproject.org, R core team, 2020).

**3 Results**

**3.1 Water chemistry**

Seawater carbonate chemistry (Table 1) differed significantly between samples. Obtained ranges were as follows: pH = 7.2–8.1 , $\Omega_{calcite}$ = 1.3–6.2, $[CO_3^{2-}]$ = 52–240 µmol kg$^{-1}$, $T_A$ = 2044-3108 µmol kg$^{-1}$, and $C_T$ = 1725–3197 µmol kg$^{-1}$. The temperature was similar between sites ranging from 26.1 to 27.9, while salinity decreased with proximity to the springs, ranging from 37 to 28. The BIOENV analysis and global

BEST test revealed that the best combination (p-value = 0.01) of environmental variables with species abundance was observed when considering pH, $[CO_3^{2-}]$, $\Omega_{calcite}$ and T ($\rho$ = 0.55), in which $CO_3^{2-}$ and pH were the environmental variables resulting in the best correlation ($\rho$ = 0.5) and salinity ($\rho$ = 0.33) and temperature ($\rho$ = 0.038) the lowest. Concerning the taxonomic metrics, the multiple regression analysis presented similar results. For diversity, richness, and evenness the pH model presented the lowest AIC

value (26.96, 196.65, -67,05, respectively), indicating the central influence of this variable on the communities, while salinity (43.77, 209.71, -54,79, respectively) and temperature (59.46, 224.28, -40.65, respectively) were less influential. Interestingly, $T_A$ and $C_T$ were the most important variables correlating to foraminiferal density (AIC = 401.79, and 401.99). The salinity (AIC = 406.34) and temperature (AIC = 409.03) were not significant variables affecting foraminifera density. Weighing by relative likelihood

(Akaike weights), log-likelihood, significance and level of variation explained by each of the environmental parameters ($R^2$) are available in Table S1. Considering its predominant influence, pH will be used as the primary variable for discussion of the potential impacts of changing carbonate chemistry. We note however that pH, $\Omega_{calcite}$, $T_A$, $[CO_3^{2-}]$ and $C_T$ are all strongly related to each other as important components of carbonate chemistry.

**Table 1** Carbonate chemistry parameters of discrete water samples collected near the substrate at the time of sediment collection. $T_A$ = total alkalinity; $C_T$ = total inorganic carbon; $[CO_3^{2-}]$ = carbonate ion concentration; $\Omega_{calcite}$ = saturation state with respect to calcite; T = temperature.

| Site | Depth (m) | Distance | $A_T$ (μmol kg$^{-1}$) | $C_T$ (μmol kg$^{-1}$) | pH | $CO_3^{2-}$ (μmol kg$^{-1}$) | Ω Calcite | T (C°) | Salinity |
|------|-----------|----------|------------------------|------------------------|-----|------------------------------|-----------|--------|----------|
| Norte | 5.8 | Center | 2611 | 2588 | 7.38 | 67.03 | 1.66 | 27.5 | 32.21 |
| | | 25 cm | 2734 | 2734 | 7.34 | 60.93 | 1.53 | 27.2 | 30.70 |
| | | 50 cm | 2699 | 2694 | 7.34 | 62.20 | 1.54 | 27.2 | 31.90 |
| | | 1 m | 2451 | 2314 | 7.66 | 118.47 | 2.85 | 27.0 | 35.25 |
| Pargos | 6.8 | Center | 3000 | 3048 | 7.23 | 52.73 | 1.33 | 27.6 | 29.95 |
| | | 25 cm | 3054 | 3047 | 7.38 | 71.16 | 1.82 | 27.7 | 28.00 |
| | | 50 cm | 2304 | 2160 | 7.72 | 119.78 | 2.97 | 27.6 | 32.00 |
| | | 1 m | 2387 | 2084 | 8.00 | 220.39 | 5.36 | 27.5 | 34.20 |
| | | > 1 m | 2336 | 2012 | 8.01 | 229.56 | 5.49 | 27.6 | 36.17 |
| Gorgos | 7.2 | 25 cm | 2350 | 2065 | 7.98 | 207.09 | 5.03 | 27.3 | 34.40 |
| | | 50 cm | 2364 | 2004 | 8.10 | 255.79 | 6.18 | 26.8 | 34.80 |
| | | 1 m | 2044 | 1725 | 8.09 | 216.08 | 5.24 | 26.9 | 34.40 |
| | | > 1 m | 2325 | 2033 | 7.96 | 209.44 | 5.02 | 27.8 | 35.90 |
| Laja | 5.8 | Center | 2827 | 2756 | 7.51 | 102.65 | 2.50 | 27.9 | 32.75 |
| | | 25 cm | 2590 | 2385 | 7.83 | 164.17 | 4.00 | 26.1 | 33.70 |
| | | 50 cm | 2354 | 2013 | 8.05 | 240.04 | 5.70 | 26.4 | 36.70 |
| | | 1 m | 2319 | 2051 | 7.94 | 192.93 | 4.59 | 26.5 | 36.60 |
| | | > 1 m | 2357 | 2092 | 7.90 | 193.55 | 4.63 | 28.1 | 36.17 |
| Agua | 5.4 | Center | 2444 | 2167 | 7.93 | 203.84 | 4.90 | 27.4 | 35.60 |
| | | 25 cm | 2364 | 2128 | 7.87 | 176.51 | 4.27 | 28.0 | 35.10 |
| | | 50 cm | 2314 | 2088 | 7.85 | 168.22 | 4.07 | 28.4 | 35.10 |
| | | 1 m | 2347 | 2063 | 7.95 | 206.13 | 4.98 | 28.2 | 35.10 |
| | | > 1 m | 2363 | 2049 | 8.01 | 226.08 | 5.47 | 27.7 | 34.90 |
| Mini | 4.9 | 25 cm | 2443 | 2071 | 8.08 | 265.01 | 6.31 | 26.9 | 36.50 |
| | | 1 m | 2365 | 2113 | 7.90 | 184.16 | 4.37 | 26.6 | 36.90 |
| | | > 1 m | 2356 | 2049 | 7.99 | 218.13 | 5.16 | 26.4 | 37.30 |

**3.3 Taphonomy and assemblage test size analysis**

Along the gradient of changing carbonate chemistry, a significant change in foraminiferal test size was observed (Kruskal-Wallis, chi-squared = 16, df = 3, p-value = ≤ 0.01). A gradual decrease in the abundance of tests with smaller surface area and a relative increase of larger tests are observed towards low-pH sites (Fig. 2a, $R^2$ = 0.73, p-value = ≤ 0.01), whereas an abrupt increase in test size was observed at 7.8 pH . The post hoc Dunn's test reveals that only differences between present-day and extremely low-pH conditions, which are beyond the predicted to the end of the 21$^{st}$ century was significant (z = -2.7, p-value = ≤ 0.01). Specifically, average test size in the assemblage more than tripled when compared to present-day conditions (from $0.33 \pm 0.2$ to $0.87 \pm 0.14$ mm$^2$). This large change can be visualized in Fig. 2a, and is likely related to changes in faunal composition rather than interspecific changes in species size. As observed in the taphonomy analysis, linear correlation with respect to the dominant taxa coverage, i.e., the species *A. angulatus,* shows a high and significant correlation to changes in average assemblage test size (Fig. 2c, $R^2$ = 0.89, p-value = ≤ 0.01). Raw data of assemblage average test size, and taxonomic metrics are available in table S2.

Sites with highest pH (~8.1 pH ) at PM are relatively pristine with well-preserved tests representing approximately 80% of the assemblage, however, this gradually changes as the effects of spring water increase (Fig. 2b). In general, dissolution was not homogenous between species, but mainly associated with the occurrence of LBF, specifically, *Archaias angulatus,* which was able to individually explain 73% of highly dissolved tests occurrence ($R^2$ = 0.73, Fig. 2c). The small, less robust calcifiers (e.g., *Rosalina* spp., *Elphidium* spp.) were rare but when found they were mostly in pristine conditions. Regarding color patterns (table S2), only two specimens with brown color were found at spring Laja, indicating little reworking of sediments and therefore, mixing of pristine and relict tests. Overall, we observed that the specimens were in good conditions, composed of well-preserved time-averaged assemblages, which thus provide a good representation of the present-day biocoenosis (Yordanova and Hohenegger, 2002). However, at 7.7 pH and lower, high levels of taphonomical alteration started to occur (Fig. 2b-c, dashed lines), when poorly preserved tests comprised ~ 50 % of total assemblage.

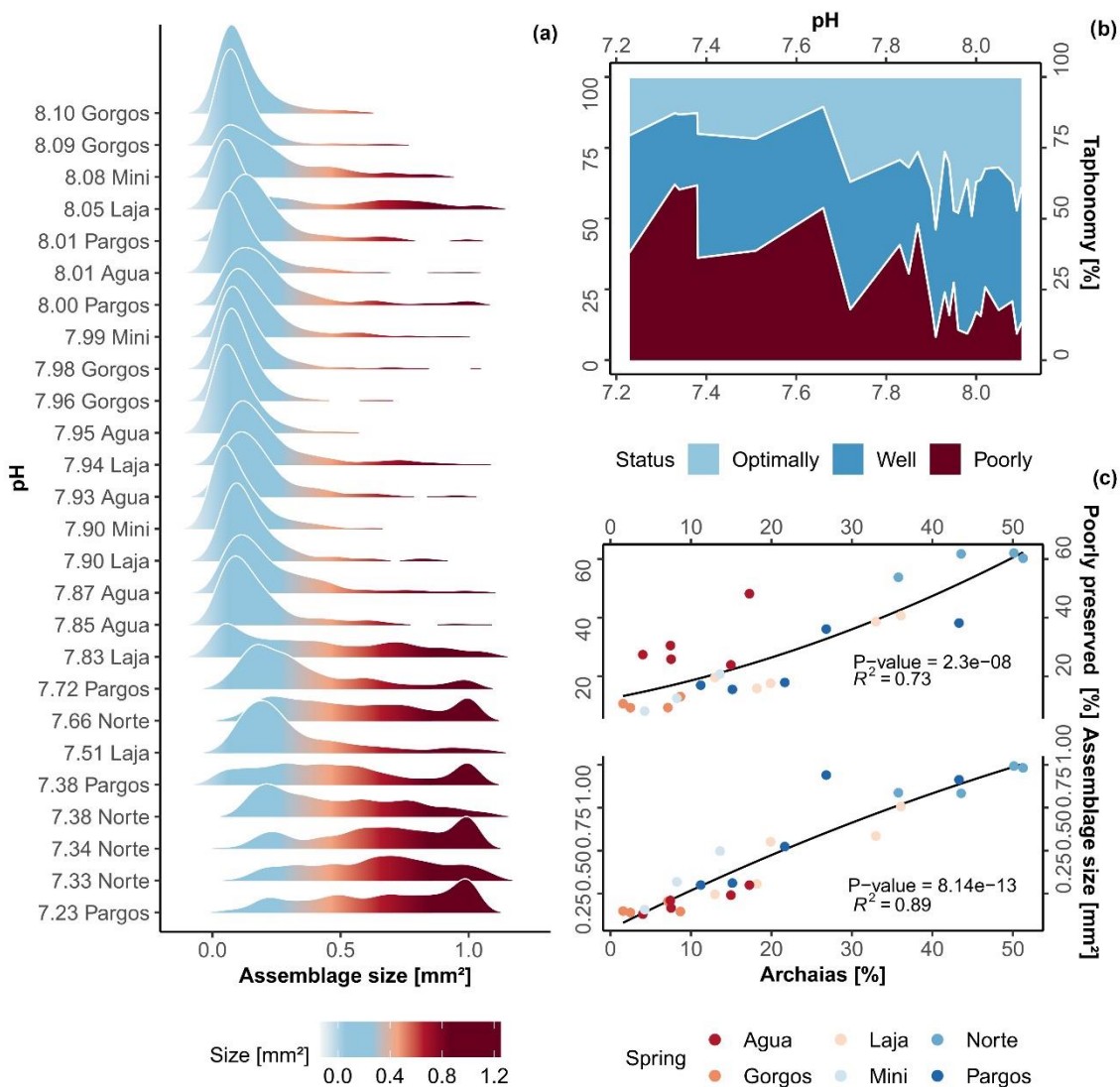

**Figure 2** (a) Density plot of assemblage test size, (b) area plot of foraminifera taphonomical status against pH, and (c) variation of poorly preserved tests and average assemblage test size against *Archaias angulatus* relative contribution. The black lines represent the second-order polynomial model fits along with the $R^2$ value and p-value (c).

### 3.2 Foraminiferal analysis

The assemblages (live + dead, table S3) found at PM exhibit similar composition to previous studies conducted in nearby coastal settings (Gischler and Möder, 2009), Caribbean eastern islands (Wilson and Wilson, 2011), and the Gulf of Mexico (Stephenson, Hallock and Kelmo, 2015; Amergian et al., 2022). A total of 8564 foraminifera from 141 species were identified, belonging to 4 orders, 37 families, and 73 genera. Agglutinated species contributed 6.4 % (9 species), porcelaneous 61 % (86 species), and hyaline 32.6 % (45 species) of the total species richness. For total assemblages the species *A. angulatus* (9.4 %), followed by *Rotorbinella rosea* (9.3 %), *Asterigerina carinata* (6.9 %), and the *Rotorbis auberii* (4.7 %)

were the most important contributing taxa, whereas for living counts *Rosalina globularis* was the most important taxa (11 %). Species that contributed at least 3 % of total abundance are shown in Fig. 3.

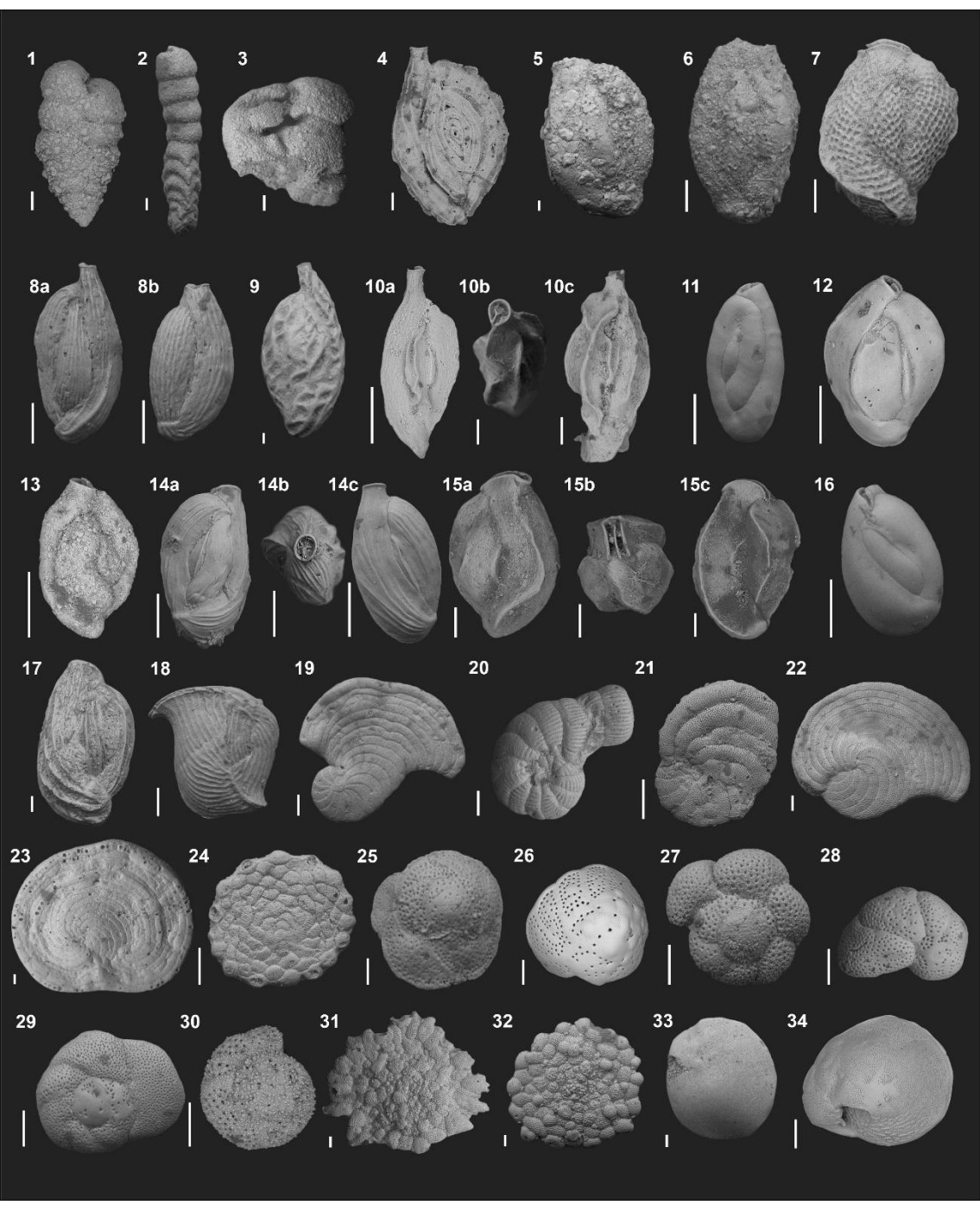

**Figure 3** Electron micrographs of foraminifera species from Puerto Morelos reef lagoon springs constituting at least 3% of the assemblage. All scale bars represent 100 µm. *1 Textularia agglutinans*, lateral view. *2 Clavulina angularis*, lateral view. *3 Valvulina oviedoiana*, lateral view. *4 Spiroloculina corrugata*, lateral view. *5 Agglutinella compressa*, lateral view. *6 Schlumbergerina alveoliniformis*, lateral view. *7 Lachlanella carinata*, lateral view. *8 Quinqueloculina subpoeyana*, lateral view. *9 Quinqueloculina*

*tricarinata*, lateral view. **10a, 10c** *Quinqueloculina* conf. *Quinqueloculina distorqueata*, lateral views. **10b** *Quinqueloculina* conf. *Quinqueloculina distorqueata*, apertural view. **11** *Quinqueloculina bosciana*, lateral view. **12** *Quinqueloculina disparilis*, lateral view. **13** *Quinqueloculina* conf. Q. *berthelotiana*, lateral view.

**14a,14c** *Quinqueloculina carinatastriata*, lateral views. **14b** *Quinqueloculina carinatastriata*, apertural view. **15a,15c** *Affinetrina quadrilateralis*, apertural views. **15b** *Affinetrina quadrilateralis*, apertural view. **16** *Miliolinella elongata*, lateral view. **17** *Pseudotriloculina linneiana*, lateral view. **18** *Articulina pacifica*, lateral view. **19** *Laevipeneroplis proteus*, lateral view. **20** *Peneroplis pertustus*, lateral view. **21** *Peneroplis planatus*, lateral view. **22** *Archaias angulatus*, lateral view. **23** *Cyclorbiculina compressa*, lateral view. **24** *Sorites marginalis*, lateral view. **25** *Rotorbis auberii*, spiral view. **26** *Rotorbinella rosea*, spiral view. **27** *Trochulina* sp, spiral view. **28** *Rosalina* cf. *floridana*, spiral view. **29** *Rosalina globularis*, spiral view. **30** *Cibicidoides* sp, spiral view. **31** *Planorbulina mediterranensis*, lateral view. **32** *Planogypsina acervalis*, lateral view. **33** *Amphistegina gibbosa*, lateral view. **34** *Asterigerina carinata*, lateral view.

In general, the species *Quinqueloculina tricarinata*, *A. angulatus*, *Amphistegina gibbosa*, *Valvulina oviedoiana, Ciclorbiculina compressa* increased towards low-pH, high $C_T$, and high $T_A$ values, presenting an increased relative abundance and lower sensitivity to OA. On the contrary, highly sensitive species include *Thochulina* sp., *Sorites marginalis*, *Quinqueloculina subpoeyana*, *R. auberii*. The species *Rotorbinella rosea*, *Clavulina angularis*, *Quinqueloculina disparilis*, *Lachlanella carinata,* and *Schlumbergerina alveoliniformis* decrease in abundance towards low pH at a lower rate compared to the highly sensitive species suggesting higher tolerance to lower saturation states.

The symbiont-bearing taxa (Fig. 4a, $R^2 = 0.59$, p-value = $\leq 0.01$) presented lower sensitivity to OA conditions, increasing in relative abundance towards low-pH. The small miliolids (Fig. 4c, $R^2 = 0.42$, p-value = $\leq 0.01$), opportunistic (Fig. 4d, $R^2 = 0.28$, p-value = $\leq 0.01$), and small rotaliid taxa (Fig. 4e, $R^2 = 0.36$, p-value = $\leq 0.01$) decreased in relative abundance towards low-pH conditions, indicating higher sensitivity. Kruskal-Wallis analysis reveal that the observed variation was statistically significant for most functional groups: Symbiont-bearing (chi-squared = 13, df = 3, p-value = $\leq 0.01$), small miliolids (chi-squared = 12, df = 3, p-value = $\leq 0.01$), opportunistic (chi-squared = 16, df = 3, p-value = $\leq 0.01$), and small rotaliids (chi-squared = 9, df = 3, p-value = $\leq 0.01$). Post hoc Dunn test reveals that significant changes occurred predominantly between present day (~ 8.1 pH ) and extremely low-pH conditions ($\leq 7.6$ pH ) representing conditions beyond those predicted for the end of the 21$^{st}$ century: Symbiont-bearing (z = −2.38, p-value = 0.01), small miliolids (z = 2.7, p-value = $\leq 0.01$), and opportunistic (z= 2.4, p-value = 0.01). For small rotaliid taxa the significance was observed between low-intermediate acidification scenarios (~7.9 pH ), at which the group presented a higher contribution, and extremely low-pH conditions ($\leq 7.6$ pH ) where a strong decrease was observed (z = 1.7, p-value = $\leq 0.01$). No significance in abundance between sites was observed for agglutinated foraminifera (chi-squared = 2, df = 3, p-value = 0.5), which also did not present significant correlation with changing pH (Fig. 4b, $R^2 = 0.11$, p-value = 0.1). Raw data of functional

and test type group are given in table S4 and the distribution of functional groups against changing pH in Fig. 4.

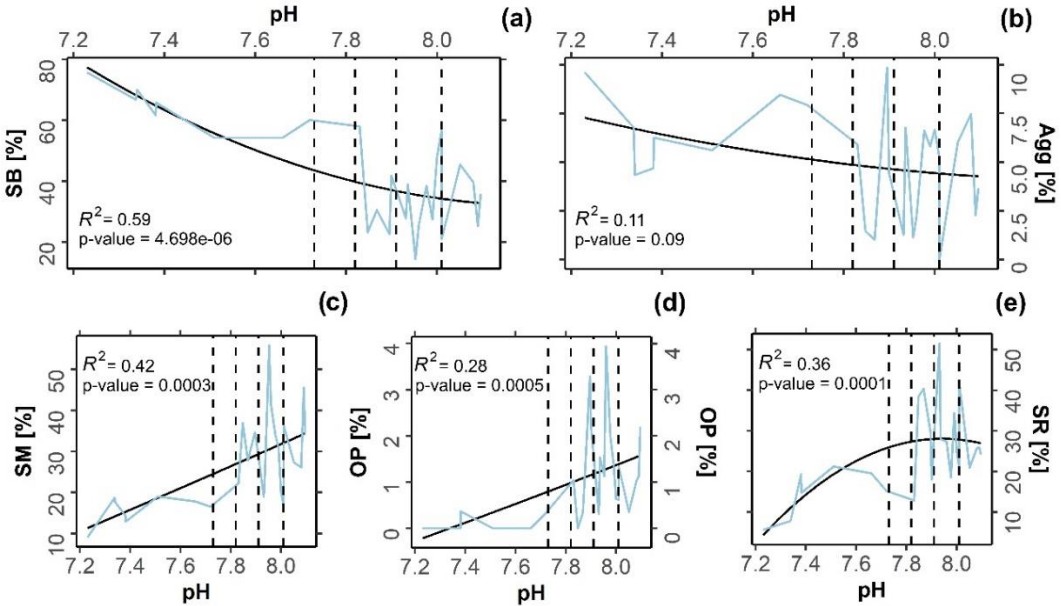

**Figure 4** Variation of functional groups versus pH. The black line represents the second-order polynomial model fits along with the $R^2$ value and the blue line represents the raw values obtained from in situ assemblages. Dashed lines demark predicted pH values at the end of this century following the Coupled Model Intercomparison Project Phase Six (CMIP6) predictions for Shared Socioeconomic Pathways (SSP1-2.6: 8.01 pH ; SSP2-4.5: 7.91 pH ; SSP3-7.0: 7.82 pH , and SSP4: 7.73 pH ). SB = symbiont bearing, Agg = agglutinated, SM = small miliolids, OP = opportunistic, SR = small rotaliids.

All taxonomic metrics presented a gradual decrease towards the springs (Fig.5a-d). On average, H′ ranged from 3.9 to 1.6 (Fig. 5a, $R^2 = 0.72$, p-value = 4.8$^{-08}$); S from 71 to 11 (Fig. 5b, $R^2 = 0.67$, p-value = 3.3$^{-07}$); J′ from 0.9 to 0.6 (Fig. 5c, $R^2 = 0.64$, p-value = 9.5$^{-07}$), and foraminiferal density from 2167 to 36 ind./cm$^3$ (Fig. 5d, $R^2 = 0.22$, p-value = 0.02). Kruskal-Wallis analysis revealed that the observed variation was statistically significant for all taxonomic metrics: N (chi-squared = 14.5, df = 3, p-value = ≤ 0.01), S (chi-squared = 20, df = 3, p-value = ≤ 0.01), J′ (chi-squared = 15, df = 3, p-value = ≤ 0.01) and H (chi-squared = 19, df = 3, p-value = ≤ 0.01). However as observed for functional and test type groups, the post hoc Dunn test revealed that changes were most significant between present day and extremely low-pH conditions: N (z = 2.2, p-value = 0.02), S (z = 3.4, p-value = ≤ 0.01), J (z = 3.1, p-value = ≤ 0.01), and H (z= 3.4, p-value = ≤ 0.01). No significant difference was observed for any taxonomic metric at low-intermediate acidification scenarios (SSP1-2.6 and SSP2-4.5), and only S differed significantly between present day and high acidification scenarios (Fig. 5b, SSP3-7.0 and SSP5-8.5, z = 2.1, p-value = 0.03).

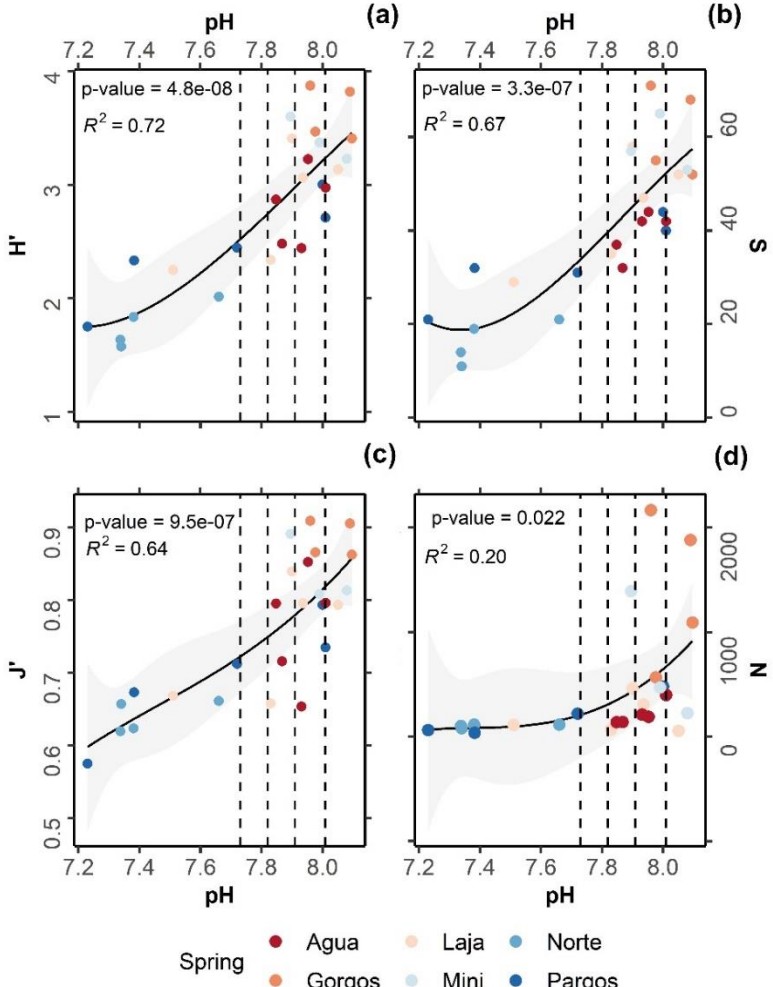

**Figure 5** Relationships between pH and (a) Shannon-Weiner Diversity Index (H'), (b) species richness (S), (c) Pielou's evenness (J'), and (d) foraminiferal density (N). The black lines represent second-order polynomial model fits, and grey areas mark 95 % confidence intervals. Dashed lines demark predicted pH values at the end of this century following the Coupled Model Intercomparison Project Phase Six (CMIP6) predictions for Shared Socioeconomic Pathways (SSP1-2.6: 8.01 pH ; SSP2-4.5: 7.91 pH ; SSP3-7.0: 7.82 pH , and SSP4: 7.73 pH ).

Data analyses indicates that under the most conservative projections (SSP1-2.6; SSP2-4.5) foraminiferal assemblages did not display considerable changes in taxonomic metrics, relative to assemblages living at present-day conditions. For projections SSP3-7.0 and SSP5-8.5 the analyzed assemblages presented a significant decrease in richness S, indicating that foraminifera assemblages are likely to be affected under high acidification scenarios. At the species level, agglutinated foraminifera were not measurably influenced by changes in pH, while the small rotaliids and symbiont-bearing taxa presented relatively higher resistance. For conditions beyond those predicted for the late 21st century, foraminifera density decreased abruptly and high taphonomical alteration was observed.

### 3.4 X-ray MicroCT

The X-ray MicroCT (Fig. 6a-d) analysis revealed that despite having a similar size ($0.80 \pm 0.05$ mm$^3$) and

volume ($0.06 \pm 0.02$ mm$^3$), specimens present at low-pH conditions (7.1 pH ) were on average 46 % less

dense ($2.4 \pm 0.2$ to $1.30 \pm 0.03$ g/cm$^3$) than the specimens present at high-pH conditions (Welch Two Sample

t-test, t = 8.1204, df = 3.0808, p-value = 0.0035). Yet, no significant (Two Sample t-test, t = -1.4378, df =

6, p-value = 0.2) difference in chamber wall thickness was observed ($0.050 \pm 0.006$ mm). The differences

in internal density (Fig. 6a and 6b) represent 2 specimens living in high and low-pH conditions,

respectively. The external differences of these same individuals are represented in the 3D volume image in

Fig. 6c and 6d. Raw data of test density, chamber wall thickness, test volume, and test diameter measured

in *A. angulatus* individuals are listed in table S5.

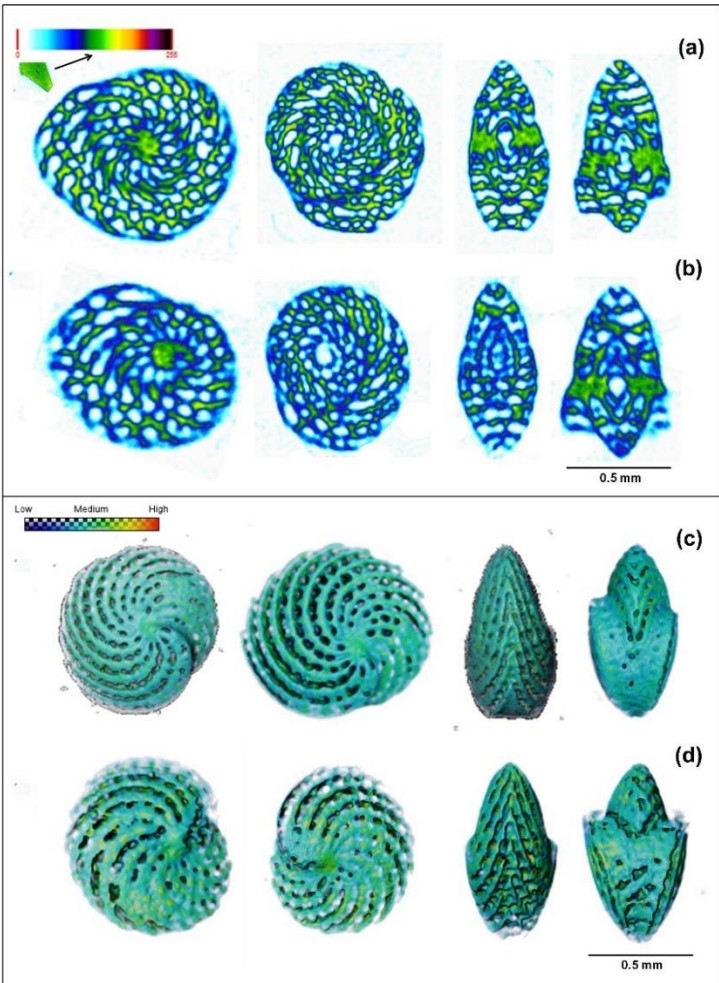

**Figure 6** - Comparison between X-ray microCT images with color code as a function of calcite density.

The specimen living at ~ 7.96 pH  (a) presents a higher calcite density (greener) when compared with low

~ pH 7.11 individual (bluer) (b). The 3D volume rendering in function of calcite density for the same

individuals living at the high (c) and low-pH conditions (d). Note that the individual at "d" living under

low pH presents a test with incomplete parts and blurred edges, which demonstrates a lower density.

### 4 Discussion

## 4.1 Foraminiferal resistance to intermediate pH conditions

Under the two most conservative acidification projections (Fig. 5a-d) foraminifera assemblages in PM did not display considerable changes, while at high acidification scenarios a significant decrease in species richness was observed. These results indicate that benthic foraminifera are unlikely to be affected by pH decreases of ~ 0.2, but certainly respond adversely to higher acidification levels (~ 0.4 pH ). These findings are generally consistent with previous observations from other naturally high $p$CO$_2$ sites in which taxonomic metrics decreased significantly with declining pH (Bernhard et al. 2009; Dias et al. 2010; Pettit et al., 2015; Dong et al., 2019, 2020). It is noteworthy, however that changes in assemblage composition did not follow the same pattern observed in these previous studies. Whereas the proportion of calcareous species usually decline with decreasing pH, they remained dominant in PM (~90 %, mainly SB, Fig. 4a) under all projections.

Considering the mid-range pH (~ 7.9 pH ), small rotaliids are more resilient (Fig. 4e); the chemical conditions at PM, along with the physiology of calcification in foraminifera may explain the lack of sensitivity of these species. Recent calcification models demonstrate that hyaline foraminifera can manipulate pH to control the speciation of inorganic carbon parameters during calcification (De Nooijer et al., 2009; Toyofuku et al., 2017; De Goeyse et al., 2021; Geerken et al., 2022). Specifically, the proton-pumping based model (Toyofuku et al., 2017) shows that a decrease in pH (~ 6.9 pH ) in the environment induces the transformation of $CO_3^{2-}$ and bicarbonate (HCO$^{3-}$) into CO$_2$, whereas at the site of calcification the elevated pH (~ 9 pH ) results in the opposite shift  converting to $CO_3^{2-}$. As foraminifera induce pH changes exceeding those predicted from SSP1-2.6 and SSP2-4.5, intermediate acidification scenarios are unlikely to impair foraminiferal calcification. In fact, the higher abundance of small rotaliid (Fig. 4e) and resistance of SB genera (e.g., *Amphistegina*) supports the hypothesis that they might, at least to a certain extent, benefit from the extra dissolved inorganic carbon (Toyofuku et al., 2017, De Goeyse et al. 2021). This interpretation is consistent with the observation that carbonic anhydrase plays a key role in the biomineralization process of some rotaliids, possibly concentrating inorganic carbon for calcification by converting HCO$_3^-$ into CO$_2$ (De Goeyse et al. 2021). As such, these models suggest that increased CO$_2$ might favor foraminifera calcification by increasing C$_T$, which is notably higher towards the springs in PM (Table 1).

The higher C$_T$ and T$_A$ may also induce CO$_2$ fertilization effects in SB species, increasing the activity of symbionts (Fujita et al., 2011; Uthicke and Fabricius, 2012; Martinez et al., 2018). This could explain why symbiont-bearing species including *A. angulatus* (chlorophyte-bearing), increased in relative abundance from 11-15 % to 21 % from ~ 8 to 7.72 pH  (e.g., Pargos spring). This behavior was also observed for *Amphistegina gibbosa* (diatom-bearing), increasing from 16-19 % to 23 %, and *Cyclorbiculina compressa* (chlorophyte-bearing) that contributed only ~ 1 % at high-pH stations but increased in relative contribution to 3.2 % at intermediate pH.   These findings are supported by laboratory-controlled experiments demonstrating that both *A. angulatus* (Stuhr et al., 2021) and *Amphistegina* sp. (McIntyre-Wressnig et al., 2013; Prazeres et al., 2015) can calcify and live under relatively low-pH conditions (~ 7.6 pH ).

This behavior, however, was not observed for all SB. For example, the species *H. depressa* was documented to be resilient in laboratory-controlled conditions (Vogel and Uthicke et al., 2012; Schmidt, Kucera and

Uthicke, 2014), but in PM it showed a strong decline towards low-pH waters. A possible explanation is that
although salinity is considered to not be a significant factor controlling the overall foraminiferal communities (AIC, BIO-ENV/global BEST analysis), salinity may have specifically affected the occurrence of select stenohaline (30–45) species, e.g., larger rotaliids (Hallock, 1986). However, this is unlikely important since salinity at the springs is > 30 over 90 % of the time and does not drop below 27 (Crook et al., 2012; Martinez et al., 2018). Other parameters, such as heavy metals could also influence the abundance of certain species, but concentrations of the metals (Paytan, unpublished) were not significantly higher at the springs when compared to sites not influenced by spring water discharge (> 1m away from the discharge sites). Hence, we do not attribute the changes in foraminifera assemblages to impacts of heavy metals. We note that the all of the samples collected and analyzed here are just a few meters apart hence other parameters such as light, eutrophication, and pollution are identical.

The high $C_T$ and $T_A$ might also raise local pH and carbonate saturation during photosynthesis, even if only on the specimen-scale (i.e. at the foraminiferal shell surface). A diffusive boundary layer of high pH (up to 8.9) has been documented at the underlying surface of symbiont bearing foraminifera (Koehler-Rink and Kuehl, 2000; Glas et al., 2012), and although insufficient to compensate future decreases of ambient seawater pH, it might increase the symbiont-bearing species resistance against (periodical) lowered saturation states. Correspondingly, the symbioses between seagrasses and foraminifera has also been suggested as a key factor in the resilience of epiphytic species (e.g., *A. angulatus*, *C. compressa* and *A. gibbosa*). Although no significant effect has been reported for some species (Fabricius et al., 2011; Pettit et al., 2015), *Marginopora vertebralis* was observed to maintain its growth when associated with its common algal host, *Laurencia intricata* in laboratory conditions (Doo et al., 2020). With respect to observations in the present study, the epiphytic species *R. globularis* was the abundant taxon in the living community (11 %), and although not the primary objective of the present study, it gives important insights about short-term foraminifera responses. Specifically, this finding agrees with the observed resilient behavior of Rosalinids in the natural, low-pH venting sites of Panarea (Di Bella et al., 2022). Considering the low occurrence of fully stained tests, future analysis on phytal substrates in PM would be necessary to confirm this trend. Lastly, the ability of foraminifera to function and calcify near the springs may also be related to the site-specific natural pH variability to which the species are exposed. For many coastal/transitional areas characterized by high $pCO_2$ variability, foraminifera seem to be more resilient and acclimated to changing conditions including low pH (Haynert et al., 2012; Charrieau et al., 2018). By discharging low-pH waters for millennia (Back et al., 1979) the foraminifera living near the spring have experienced pH variability over a much longer timespan than the life span of individual organisms (Martinez et al., 2018). Specifically, as reef-dwelling organisms, the foraminifera in PM experience a wide range of pH on daily and seasonal scales which might physiologically increase the species resilience at low-pH conditions (Price et al., 2012).

**4.2 High acidification scenarios**

Previous data from recruitment and succession experiments in PM showed that foraminifera were able to calcify and increased in weight over the investigated period (14 months) at low (~ 7.8) pH conditions (data from Laja and Gorgos springs, Crook et al., 2016). Two years later, Martinez et al. (2018) documented the occurrence of calcareous tests at PM even at extreme acidification levels (~ 7.1 pH ). In agreement, we

observed that despite the strong decrease in foraminifera density calcareous foraminifera remained dominant in PM sediments (~90 %, mainly *A. angulatus* and *A. gibbosa*) even at expected future conditions for the end of the 21st century and beyond.

For high acidification scenarios (SSP3-7.0 and SSP5-8.5), the in-situ occurrence of calcifying foraminifera has only been reported in the deep-sea near extensive $CO_2$ vents in the Wagner Basin (Pettit et al., 2013). At this site, a rich food supply and stable temperatures were hypothesized to compensate the effects of OA and explain the shift towards opportunistic assemblages. The springs from PM also have relatively high nutrient concentrations compared to the open waters in the region (Null et al., 2014; Crook et al., 2016), however, near the springs, assemblages did not change towards opportunistic dominated assemblages, suggesting that the nutrient availability does not exert a major control at this site. Rather, the high-pH assemblages which was heavily dominated by small calcareous forms were replaced by larger symbiont-bearing species near the springs (Fig. 4a-e). Such species are known to be sensitive to high nutrient loading, likely because of changes in turbidity/light regimes and their dependence on algal symbionts to enhance growth and calcification (Hallock et al., 2003; Prazeres et al., 2020; Girard et al., 2022). However, at PM despite higher nutrient levels the waters at the springs are clear and light regimes are not reduced (water depth at the spring sites is 5-7 meters).

To explore the causes for the resilience seen in certain taxa (Fig. 4a-d) and investigate possible acclimation patterns, we employed an X-ray microCT analysis in *A. angulatus* specimens from high and low pH conditions. The analysis (Fig. 6a-d) revealed that despite having similar size, volume, and chamber wall thickness  the specimens found at low-pH conditions (7.1 pH ) were on average 46 % less dense than the specimens present at high-pH conditions (7.96 pH ). This demonstrates that this species is able to calcify in low-pH conditions beyond those predicted for the late 21st century albeit with shells that have a lower average density. This indicates that *Archaias* individuals were not capable to acclimate sufficiently to maintain calcification efficiency similar to ambient present-day rates. These results agree with Knorr et al. (2015) who observed a 50 % decrease in *A. angulatus* size at 7.6 pH , and a consequent decrease of 85 % in the production of high-Mg calcite by this species, and also with other published results for SB species such as *Peneroplis* spp (pH 7.4, approximately 25 % lower, Charrieau et al., 2022), and *Amphistegina* spp (pH 7.6,  approximately 20 % lower, Prazeres et al., 2015). We acknowledge that *post-mortem* dissolution may also contribute to the observed lower density, but only the most pristine tests were analyzed, so this influence must be minimal. Future analysis of B isotopes and B/Ca ratios could provide more information about the trends documented in the present study.  Since *A. angulatus* showed lower density close to the low-pH springs and hence is negatively impacted by the low-pH, the species increase in relative abundance towards the springs is probably associated with the high preservation potential of its tests. The tests of *A. angulatus* are large, thick, and reinforced by internal partitions (pillars), therefore more likely to be preserved in the sediment (Martin, 1986; Cottey and Hallock, 1988). This is confirmed by the performed regression analysis as the species relative contribution explains 88 % of assemblage test size and 73 % of high dissolved test occurrence in the samples (Fig. 2c). In fact, changes were so abrupt that shifts in the assemblage test size and functional groups were clearly observed at ~7.7 pH  (Fig. 2a), when the symbiont-bearing taxa relative contribution also started to increase (Fig. 4a). At this point preservation thresholds of

smaller taxa seemed to be crossed, and their decrease in relative abundance near the springs is likely related to higher rates of breakage and dissolution (Present study, Martinez et al., 2018).

Considering foraminifera a crucial component of reef sediment production (Langer et al., 1997; Langer, 2008), including *A. angulatus* in the Caribbean region, our results support previous findings that reef-building carbonate production and accumulation are likely to decrease under future OA scenarios, even in the tropics (Knorr et al., 2015; Eyre et al., 2018; Kuroyanagi et al., 2021). Specifically, we also observed a decrease in foraminifera test density (Fig.5d) and therefore in carbonate accumulation. As OA intensifies, symbiont-bearing taxa, which demonstrated higher resistance to low-pH (> 7.8 pH ), will likely still represent major contributors in the Caribbean and Gulf of Mexico sediments where species like *A. angulatus* may dominate (Culver and Buzas, 1982). In contrast, the high sensitivity of *Quinqueloculina* spp., *Triloculina* spp., *Articulina* spp., and *Miliolinella* spp. to low pH highlighted their lower fitness in response to OA, demonstrating that changes in abundance of small taxa can be used as bioindicators to monitor the effects of OA.

The relative contribution of agglutinated foraminifera slightly increased towards the low pH springs (Fig. 4b), but they did not fully compensate the decline in calcareous species (Fig. 4a–e). Since the particles available for the agglutinated tests in this region are made of calcium carbonate and under low-$\Omega$ waters these particles are also prone to dissolution possibly affecting agglutinating species. Interestingly, agglutinated foraminifera also presented species-specific responses to acidification like calcareous foraminifera. For example, *Valvulina oviedoiana* increased in relative abundance towards low pH, while *Textularia agglutinans* presented a strong decrease. Since acidification is expected to have little direct effect on agglutinated foraminifera the observed interspecific behavior is also probably associated with preservation potential. The variation of agglutinating material (e.g., mucopolysaccharide), structure (e.g., fibrous, strands, foam-like masses), and size of granular particles (e.g., fine, and coarser) are essential to determine the preservation and accumulation of agglutinated tests (Bender and Hemleben, 1988). The most important agglutinated species, in our study e.g., *T. agglutinans*, *C. angulata,* and *V. oviedoiana* use calcite cement as the binding material of particles, which probably results in a higher resistance to dissolution (Bender, 1995). Among these, *T. agglutinans* lower resistance likely responds to its smaller size, which enhances dissolution (Bender, 1995). Altogether, we observe that between 8-1 and 7.8 pH  foraminifera physiology was a main driver of foraminifera distribution, whereas at $\leq$ 7.7 pH  (Fig. 2b) the preservation potential became an important factor affecting the distribution of both calcareous and agglutinated tests.

We cannot exclude the possibility that the higher accumulation of *A. angulatus* tests could be responsible for an overestimation in symbiont-bearing taxa density. In this case, species richness would be more reliable to the interpretation of assemblage responses, which was the only parameter to decrease significantly at a pH below 7.7  (Fig. 5b), suggesting that overall foraminifera are less likely to acclimate under high acidification scenarios. These results bring serious implications for foraminifera communities resilience in this century as SSP3-7.0 and SSP5-8.5 scenarios also predict substantial increases of sea surface te mperature (Kwiatkowski et al., 2020), which combined with surface OA might critically decrease the tolerance of foraminifera (reviewed in Kawahata et al., 2019). Recently, Bernhard et al. (2021) observed that foraminiferal assemblages presented the lowest number of species and abundances under a triple-stress

treatment (low-pH/$O_2$ and high temperature) demonstrating the synergetic effects of these variables. As observed in PM, agglutinated foraminifera were relatively more resistant than calcareous taxa.

In general, for emissions beyond the predicted to the end of 21$^{st}$ century (resulting in a pH below 7.7) all taxonomic metrics decreased significantly, and calcareous species with higher preservation potential like
*C. compressa* and *A. angulatus* comprised up to 50–60 % of the total assemblage. This was expected since a drop in the $\Omega$ aragonite < 3.2 would increase foraminifera dissolution (Yamamoto et al., 2012), but these calcareous taxa were still found at the center of discharge where the surface sediments were still composed by carbonate. We attribute this to high $T_A$ levels, which was suggested as a parameter that limits the dissolution rates of *A. angulatus* and other porcelaneous tests in the springs at the coast of Florida
(Amergian et al., 2022). The high $T_A$ may specifically provide a calcification optimum within the polyhaline (22-30) waters both at the springs in Florida and in PM where a similar range of salinity was observed. This hypothesis could explain the observed resistance of *A. angulatus* in the present study, and the higher association of foraminifera density to $T_A$. WE note that when restricting our analysis to pristine, well-preserved tests, the taxonomic metrics at 7.7-7.2 (Fig. 5) would be much lower and more like those
presented by Uthicke, Momigliano, and Fabricius (2013), in which foraminifera were almost absent at sites with ≤ 7.9 pH .

## 5 Conclusion

Despite their life-long exposure to low-pH conditions, benthic tropical foraminifera species could be negatively affected under the high acidification scenarios (SSP3-7.0 and SSP5-8.5) for the end of the 21$^{st}$
century. Species-specific responses in foraminiferal assemblages were observed and as the oceans become more acidic, reef foraminiferal assemblages may gradually shift towards hyaline, symbiont-bearing and agglutinating species. The species *A. angulatus*, which is known to be dominant in warm, oligotrophic areas of the Caribbean and the Gulf of Mexico can calcify at pH conditions lower than those projected by SSP5-8.5, however, the observed lower density of the pristine tests suggests that reef carbonate budget may
decrease as this species represents a major carbonate producer at these areas. Considering the observed trends of increasing average assemblage test size, our results demonstrate the key role smaller foraminifera have as bioindicators to monitor the effects of OA, as their high sensitivity to dissolution makes them first responders to ongoing OA.

### 6 Data availability

All data related to this study are given in the Supplement data files accompanying this paper.

### 7 Author contribution

DF, AP and CFB conceived of and designed the study. DF performed the faunal and statistical analysis. OMA and RTL conducted the Micro-CT experiments. DF, AP and CFB analyzed the data. DF, AP, CFB prepared the original draft of the manuscript with writing, and OMA and RTL reviewed and edited.

### 8 Competing interest

We declare that this manuscript has no conflict of interest.

## 9 Acknowledgments

This study was funded by the National Science Foundation-1040952 (to AP). DF thanks the scholarship of the National Council for Scientific and Technological Development (CNPq) No. 132210/2020-7. CFB acknowledges the Buzas Award for Travel (BAT) received from the Cushman Foundation for Foraminiferal Research. The funders had no role in the study design, data collection, and analysis, decision to publish, or preparation of the paper. DF thanks Pamela Hallock, Heitor Evangelista, Lennart de Nooijer, Sven Uthicke and the anonymous reviewers for their helpful comments and suggestions.

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
