# Peer review of "Acidification impacts and acclimation potential of Caribbean benthic foraminifera assemblages in naturally discharging low-pH water"

_Biogeosciences, 2022_

## Referee Comment (RC3)

Anonymous Referee#3, 9 June 2022

The present manuscript by Francois et al., 2022 presents a field study of Caribbean tropical foraminifera in the Puerto Morelos reef Lagoon. Six stations were sampled in October 2011 along a broad natural pH gradient generated by submarine springs. The study of spatial variability on foraminifera fauna driven by pH gradient is an original approach. This work shows interesting results that corroborate knowledge already known/suggested in previous studies. However, this manuscript deserves to be restructured and clarified on some major points before publication.

Major comments:

- It should be kept in mind that specimens come from the natural environment (not a controlled experiment) so multiple stresses can potentially influence calcification (salinity, eutrophication, pollution, warming...) these other parameters should be further discussed according to what is known about the site in previous studies.

- There is a need for the bibliography to be more up to date especially those published on LBF and µCT. I suggest this non-exhaustive list: Charrieau et al., (2022); Kinoshita et al., (2021); Kuroyanagi et al., (2009); Fox et al., (2020); Iwasaki et al., (2019) …

- In this paper, it is unclear about the use of live and dead fauna, if rose bengal staining has been done you must describe in the manuscript the assemblage of the foraminiferal fauna at each station and perform the ecological analyses on the live fauna. The dead fauna cannot be treated with the live fauna. If you want to study dead fauna it must be done in a separate section and clearly stated in the manuscript as "live fauna" or "dead fauna". If you study dead fauna, you must describe the assembly of dead fauna. The fact that there are few living foraminifera may be related to the seasonality of the site (previous studies?). This site may be a place of sedimentary deposition, accumulation, and currents… (1cm corresponds to what period? Previous studies?). You should discuss the sampling method used if most of the living fauna live on substrates, it could be interesting to think about a new sampling method?

- You need to clarify which data are common with the paper by Martinez et al., (2018). It seems to me that you have the same dataset or a selection of them. If you share other data from this previous paper, please indicate it clearly (this can also help to reduce the manuscript).

- Your data are related to the impacts of a natural pH gradient on a series of stations at a specific date (October 2011). You are therefore looking at spatial variability of foraminifera along a pH gradient and not at temporal variability. If you want to discuss temporal projections, I will discuss this in a discussion section. To discuss temporal projections, you need to be more nuanced because you need to know the seasonal variability of the living fauna and their interannual variability and species metabolisms (maybe you have some previous studies on this site).

- The result and discussion should be restructured, and the discussion needs major parts or titles.

Minor comments

-The title does not indicate the content of the paper I would specify LBF or tropical and the study area (to be reconsidered in the light of the new orientation of the paper)

-L41 $CaCO_3$ = calcium carbonate

-Can you clarify what you call "small or smaller foraminifera"

-L209 "live fauna" or "dead fauna"? clarify this section

-L211 Table S1 corresponds to Raw data of functional and test type groups. I think there is a file problem where is the faunal description?

-L215 Considering a 3 % contribution cutoff. Why 3%?

-L119 normally a minimum of 300 live foraminifera should be picked if the density is high

-L143 breakage and dissolution of the shells. Do you have a precise reference to do this work (quantitative approach) or is it a subjective approach?

-L154 Dissolution can affect live foraminifera and it has already been shown that some decalcified foraminifera can survive (ex. Charrieau et al 2018 Biogeosciences). To detect living foraminifera, it is either the coloration (rose bengal) or a mobility test to know if the specimen is alive or not.

-L261 p-value = 0.00 not correct (p-value < 0.001 for example)

-L165 they are many papers about CTnumber please add references

To compare µCT specimens, it is recommended to remove the ontogeny effect (growth-related), and therefore to compare the specimens they need to have the same size (standardized by the average of the maximum diameter of the individuals). It is always nice to see µCT on foraminifera, but you need to discuss that few individuals have been scanned and therefore be critical with the inter-individual variability.

-L248 For the CCA, one of the axes is not significant, it would be interesting to make an Ordistep pre-selection to select only the parameters which contribute to the CCA, and to have the two significant axes. It would be necessary to revise the design of the figure to put the variables in another color for more clarity and to put the complete legend of all the parameters used.

The legends of all figures should be complete – we need to be able to understand a figure without reading the text all the time.

-L63 Fig.4e and not 4a

-L284 add fig. 5 in the first sentence

---

## Author Response (AR1)

Referee #1: Sven Uthicke

The MS submitted Francois et al presents a field study of the effects of ocean Acidification on (sub?) tropical benthic foraminifera communities, using a unique ecosystem in Mexico as environmental proxy for high pCO2. The study finds several trends in community and species shifts of previous studies confirmed, but also detected some trends opposite to those. In general, MS is well-written and analysed. However, I have several comments (listed below) which I believe should be addressed prior to publication.

We thank the reviewer for noting that the MS is well written and provides original data.

**Major comments:**

**Comment -** My main issue is that the MS needs to discuss and investigate in more detail if other factors then pH can be the cause of community shifts or species stress. The possibility that salinity (very highly correlated with pH) may play a role is dismissed with a citation (in the results, this should be in the discussion). I think this should (and can) be further tested. For example, the linear models showing species changes along a pH axis could also be run with salinity, and with salinity and pH in the same model. Models could then be compared with AIC. Other potential influences of 'unmeasured' parameters (e.g. heavy metals) should be discussed.

**Reply:** We thank the referee for this very helpful comment. As suggested we ran the models and compared the quality of the models using the Akaike information criterion. The temperature and salinity were the models with lowest fit, even when together with pH, corroborating with the results of the BIO-ENV and global BEST test. Interestingly, we found that Alkalinity presented the best fit with foraminifera abundance, which agreed with our hypothesis that the higher $T_A$ would reduce dissolution rates and explain why calcifying forams would still occur at low-pH conditions (e.g., 7.1). The models with best fit (i.e., pH for the metrics H',J',S and $T_A$ for N) were plotted in fig. 4. The water discharging at the spring does not have elevated heavy metal concentrations hence no trend is expected since the concentrations are similar within analytical error to background ambient levels.

**Comment -** It would be good to shorten the MS, and possibly omit some sections. In fact, with some extra effort this may even constitute 2 publications. At the present state, e.g., I find the microscopy a little unconvincing, with only 4 samples from each location statistics has little power to find differences.

**Reply:** We have shortened some sections and excluded the SIMPER/Cluster analysis from the manuscript. We also relocated fig.4 (functional groups linear regressions) to the supplementary materials.

Regarding the statistics, the Kruskal-Wallis test was performed considering the entire pH gradient (regardless of the site), i.e., 26 samples for each statistical test.

**Comment -** At some places the literature does not capture all relevant studies. I think all vent studies are captured, but several experimental studies (on pCO2 effects, but also e.g. on the mentioned nutrient effects) are omitted.

**Reply:** We have re-analyzed the literature used and now added some relevant experimental studies to the manuscript. Note that there are way too many experimental studies to mention all.

**Minor comments**

**Comments**

Abstract, ln 22: shift bracket not to imply 7.1 is the end of century prediction

Introduction, Ln 30: 'to' missing in sentence?

ln 46: 'became' = 'becomes'?

ln 75 (an in other places: post mortem (2 words?)

**Reply:** We thank the referee for noticing these mistakes. We changed it accordantly.

**Comment** - throughout: pH scale needs to be specified (I assume 'seawater' or 'total'?)

**Reply:** We used seawater and it is now specified in the manuscript.

**Comment** - ln 128: if only 'total' foraminifera are used as a metric, it seems redundant to describe the staining process?

**Reply:** The description of the staining process was removed.

**Comments**

Ln 170: what are the groups the stats is conducted on for, define first

Several analyses (linear models, correlations?) are not mentioned in the stats section. Mention details of analysis (including N, error structure, random factors?) here.

**Reply:** The information is now added to the stats section.

**Comment** - Ln 237: I think you cannot call the first axis a 'gradient of acidification stress". It is a composite of several factors (including salinity)

**Reply:** We changed to a 'gradient of stress`.

**Comment** - Fig, 4,5,6: simplify and make more usual graph style. There is no need to have axis labels on all 4 sides, some labels in composite graphs can be removed.

**Reply:** We thank the referee for the suggestions and the figures were changed accordingly.

**Comment** - Fig 6 (and respective text): is this adding much, or one of those sections which could be omitted/or in another MS?

**Reply:** As explained above the simper analysis was removed from the manuscript.

**Comment** - Fig 7: also a very nice and interesting analysis, but, again, could be part of a separate MS?

**Reply:** As the community size was important for the taphonomic analysis, we decided to keep the analysis in this manuscript. That is because *Archaias angulatus* was responsible for both increasing pattens of dissolution and assemblage size due its higher preservation potential.

**Comments**

Discussion, ln 399 several experimental studies on the effect of LBF on nutrients exist.

ln 420: Kuehl et al: also a study by M. Glas. Also consider a study by S. Doo showing that LBF under OA are better off having photosynthesis or living on plants.

**Reply:** We thank the referee for the suggestions and the following studies are now cited in the manuscript:

Vogel and Uthicke, 2012, DOI: 10.1016/j.jembe.2012.05.008

McIntyre-Wressnig et al., 2012. DOI: 10.3354/meps09918

Schmidt, Kucera, and Uthicke, 2014, DOI: 10.1007/s00338-014-1151-4

Fabricius et al., 2011, DOI: 10.1038/NCLIMATE1122

Uthicke and Fabricius, 2012. DOI: 10.1111/j.1365-2486.2012.02715.x

Glas et al., 2012, DOI:10.1371/journal.pone.0050010

Doo et al., 2020, DOI : 10.1002/ece3.6552

Referee#2

We thank the referee for recognizing the importance of our work and for the useful input. Bellow, you will find the answers to the helpful comments and suggestions.

Referee#2 Major comments:

**Comment -** The paper by Francois et al. reports natural experiments of ocean acidification to understand the impact of Caribbean benthic foraminiferal assemblages near low-pH water discharging spring sites. Since this kind of natural experiments is still rare in foram research, the results are valuable and well-presented using univariate and multivariate analyses. However, I found a major issue for the authors to reconsider prior to publishing this paper.

The main issue is what time scales the authors are discussing. If the authors focus on decadal time scales occurring in this century, the conclusion in this paper is mostly incorrect.

This is because the paper deals with total (live and dead) foram assemblages in sediment and the proportion of live tests identified as stained tests is very low (3%) in the assemblages. That means that 97% of dead foram assemblages are results from long-term accumulations and taphonomic processes from various sources of habitats. Some may be in situ near sample sites, while others are **transported** or **bioturbated** particularly in shallow-water setting (the authors should show the bottom current speed and rates, as well as any benthic organisms inducing bioturbation). Some may be pristine, while others are very old (the authors may be surprised if the authors measure the radiocarbon age of foram tests). Table S3 shows that **most tests are dissolved and/or broken**. This means that dead tests are not pristine and are transported/bioturbated. Nevertheless, this paper discusses the effect of low-pH water gradient occurring in the present time as if all foram tests are in situ and very recent products.

**Reply:** We agree with the referee`s concern about the influence of post-mortem processes on our interpretations as the samples presented some alteration. However, we highlight that dissolution was not homogenous between species but mainly associated with the occurrence of LBF, specifically *Archaias angulatus* which alone was able to explain 73% of the observed dissolution (Linear regressions, $R^2 = 0.73$). The small, less robust calcifiers (e.g., *Rosalina* spp, *Elphidium* spp) were mainly pristine or in good conditions despite the expectation that these smaller species will dissolve fast and be transported by current more readily after sedimentation. We note that the focus is comparison between adjacent sites just meters apart hence they are all subjected to the same currents and bioturbation effects yet show distinct differences.

In respect to the relatively high degree of dissolution of samples as reported in Table S3, the reason for the high fraction of dissolved tests is because we simply used two general categories (1) Pristine and (2) dissolved tests, with the latter considering any degree of dissolution. To solve this issue, we have now sub-classified them as 'optimally', 'well', and 'poorly' preserved (Yordanova and Hohenegger, 2002) demonstrating that most of the tests are assigned as well or optimally preserved (at least until 7.7 pH units), indicating the overall pristine conditions of the samples.

Regarding bioturbation, we are unable to directly investigate its influence on our samples since only surface sediments (>1 cm depth) were collected. However, we analyzed test colors patterns in the forams, which can potentially be used to differentiate relict and pristine tests. The yellow, brown, or black tests reflet postmortem chemical alteration from subaerial exposure, implying a longer residence time. In contrary, the white tests would indicate a relatively little reworking of sediments. In our samples the tests were almost 100% white and translucent (we only found 2 brown specimens

in Ojo Laja), so mixing of pristine and relict tests are unlikely. Moreover, the sediment is of coarse sand size and bioturbation features such as borrows are hard to detect.

Concerning the spatial mixing, the reef lagoon circulation is not significantly affected by tides, and currents due to the microtidal regime of the region and the back-reef setting. In the springs, the waves overtopping on the reef crest and the resulting flow is the main driving mechanism of the circulation (Coronado et al. 2007, Coral Reefs, DOI: 10.1007/s00338-006-0175-9). Coronado et al. 2007 measured a relatively slow current of av. 2–3 cm s$^{-1}$, directed towards the shore at the forereef, and a faster current (av. 20 cm s$^{-1}$) was only observed through the northern and southern channels where the water exits the lagoon. Being enclosed by fringing coral reefs and not close to the exits, the currents are not considered to significantly influence the sediments distribution near the springs. Moreover, the coarse sand grain size is less conducive to dispersal by current.

To clarify the taphonomic conditions of the samples we added this discussion to the manuscript.

**Comment -** If the authors like to discuss OA impacts on foram assemblages occurring in this century, the authors should have sampled phytal and rubble substrates and studied only live assemblages. Even if the authors consider the foram tests are mostly in situ and recent products, the authors should at least show all foram taxa were found as stained tests at the sample sites.

I rather think the results are more applicable to geological OA record if the seawater chemistry of past OA records is similar to those in this study. I suggest that the authors reconsider whether your results are really able to discuss the OA impact occurring by the end of this century on living forams.

**Reply:** We highlight that the objective of the study was to assess the responses of the foraminiferal assemblages to carbonate chemistry changes on a time scale that is between decades and centuries (mid-scale), which is represented by the sediment accumulation of foraminifera tests at this site. In contrary, the study on phytal and rubble substrates would rather address the species responses within a particular point in time, as observed in previous studies (Stephenson et al. 2015, Ecological indicators, DOI: 10.1016/j.ecolind.2014.07.004).

The use of living specimens was the initial objective, but most of the tests had some significant degree of staining with Rose Bengal so it was hard to distinguish those alive from recently dead individuals. We also tried CellTracker Green that is actively taken by the cytoplasm, but only very few forams got stained so that was not useful for the statistical analysis. This pattern of low stained individuals in the sediments is common in the Caribbean and even in pristine (off-shore) reef environments (Barbosa et al., 2009, 2012, Marine Micropaleontology) as most reef-dwelling taxa tend to live on phytal or hard substrates rather than directly on the sediments. It is important to note, however, that both stained and not stained specimens were in good condition, still recording a good representation of the present-day biocoenosis (i.e., no sign of damage and aging, Yordanova, E. K. & Hohenegger, 2002), which validates the applicability of the data to investigate future impacts of OA. Additionally, the decision for total assemblages was also based on the desire to compare to Martinez et al. 2018 (Biogeosciences, DOI: 10.5194/bg-15-6819-2018) and Uthicke et al. 2013 that also used total assemblages to discuss the OA impacts occurring in this century.

**Comment -** This paper also does not discuss the possibility of dissolution of carbonate sediments including foram tests during daily pH variations (particularly during night). See the following BG paper.

https://bg.copernicus.org/articles/9/1441/2012/

**Reply:** We thank the referee for this suggestion, and we have added this relevant reference.

Referee #2: Minor comments

**Comment -** Title: be more specific (e.g. … of Caribbean benthic foraminifera to naturally discharging low-pH water)

**Reply:** We changed the title to "Acidification impacts and acclimation potential of Caribbean benthic foraminifera assemblages in naturally discharging low-pH water"

**Comment -** L42-44: this is mainly due to planktonic forams in the outer ocean.

**Reply:** We changed in the text, accordantly.

**Comments**

Table 1: add the distance from the center of spring/fracture sites, spring water flow speed and rate, and sediment grain-size distribution.

L97-98: explain in more detail; how far from the center for each site?

L97: sediment grain-size distribution data are necessary to confirm if grain size does not affect foram assemblage compositions.

**Reply:** We don't have the sediment grain-size distribution data, however, the sediment throughout the backreef lagoon is of coarse sand grain size (e.g., The beach is composed of medium carbonate sand of biogenic origin, with a mean sediment size of ~0.258 mm. Escudero et al. (2020), J. Mar. Sci. Eng.; DOI: doi.org/10.3390/jmse8040247). Moreover, considering that most species live on phytal or hard substrates (Stephenson et al. 2015) rather than directly on the sediments little or no influence of gran size is expected to our species distribution and abundance.

About the distances, the samples were retrieved at >1 m, 1 m, 50 cms, 25 cms from and at the center of discharge. This information was added in the table. The maximum distanceis 2.5 meters from the center of the discharge site.

**Comment -** L128-130: the authors should have sampled macroalge and rubble.

**Reply:** As explained above this approach wouldn't provide the information we aimed for.

**Comments**

L132: 1 ml? Is unit correct? 1 gram of sediment?

L286: ind/ml, unit correct?

**Reply:** We are now using ind/cm$^3$ as unit.

**Comment -** Fig. 3: species name should be better to express as lower-case letter to avoid confusion with environmental variables (TA vs. AT). Two ACs are found. AL>AG?

**Reply:** We thank the referee for this suggestions, the species name were changed to lower-case letter accordantly.

**Comments**

L262: Fig. 5c>4c. Check all fig numbers in this paragraph.

L263: Fig. 4a>4e

L268: H?

L276: Fig. 4e>4b

L319: table S3>S2?

L371: table S5>S4

L440: table S3>S2

L243: Thoculina>Trochulina in Fig. 2?

L725 and others: check the journal name abbreviations.

L212: Table S1 is not found in the supplementary data.

**Reply:** We thank the referee for noting these mistakes. We changed them accordingly.

**Comment -** Fig. 5: foram abundance decreased due to dissolution?

**Reply:** Yes, the samples close to the springs discharge presented lower abundance than the high pH samples. However, the density of forams was mainly explained by the alkalinity content rather than pH. We noted this correlation by comparing the linear regression models using the AIC following the suggestions of the #referee 1. As discussed in the manuscript high Alk could reduce the dissolution rates and explain the occurrence of foraminifera tests in extremely low pH conditions (e.g., 7.1 pH units).

**Comments**

Fig. 6: legends are hard to understand. What's 3% mean? Why not listed alphabetically?

Why is Similarity not listed from low to high? What does y-axis variation mean?

**Reply:** The 3% was the cut off for the analysis and the assemblages were not listed automatically used was the direct output from the software. Following the suggestions of the #referee 1 the analysis was removed.

**Comment -** L339: P-value of 0.00 is not correct expression.

**Reply:** Following the suggestions of #referee 2 and 3 we are using $p = < 0.05$.

**Comment -** Fig. 7: a) the authors should confirm if results are not affected by spring flow speed. c) the unit of assemblage test size as %? dashed lines in caption?

**Reply:** We thank the referee for noticing the mistakes in Fig.7, the unit was corrected to reflect the data and the dashed lines were added.

About the spring flow speed we don't have specific measurements from each sample station, and the temporal variability of discharge at the springs is high (impacted by tides and terrestrial recharge), however the flow is primarily in the vertical position and not laterally or towards the sediment due to buoyancy effect.

**Comment -** Fig. 8: Charrieau et al. (2022) in Sci. Rep. reports shell dissolution in living Peneroplis, the same large symbiotic miliolid forams as Archaias. They also show no significant difference in calcite density of living tests between different pH conditions. Even if the authors used pristine tests, the authors cannot tell if the calcite density changed either during living stage or post-mortem stage.

**Reply:** We have added this relevant reference. Indeed, it is impossible to tell based on our data alone if the calcite density changes were during the living stage or post-mortem. We do know that for corals at this site the low density was in the live stage. Single foraminifera analyses of B isotopes and B/Ca ratios may provide more information.

**Comments**

L381-382 may be correct for dead tests, but not for living tests.

L389 - but not living ones

**Reply:** Until 7.7 pH units the tests of foraminifera specimens still presented a good representation to biocoenosis, validly indicating the midterm responses of the local communities. Hence, we expect that this holds for both live and post-mortem as the samples include both populations.

**Comment -** L447-448: maybe due to post-mortem dissolution.

**Reply:** We added a brief discussion about a possible influence of post-mortem process to *Archaias* lower density close to the springs discharge.

**Comment -** L453: the authors cannot tell from your results.

**Reply:** This comment was removed from the manuscript. Yet considering the very high likelihood that the specimens were not transported and represent the local living assemblages the data supports the conclusion regarding calcification yet maybe not the threshold.

**Comments**

L460: relatively higher resistance of post-mortem shell dissolution to low-pH

L334-336: higher resistance to dissolution by low pH and breakage by sediment transport and bioerosion.

L468: this is what you found in your study.

**Reply:** This conclusion that SB foraminifera presented higher resistance to low-pH conditions was based in their behavior within the range of 8.1-7.8 pH units, where the dissolution isn't the main driver of foraminifera tests (as observed when pH dropped below 7.7).

**Comment** - Fig. 4: This graph shows that SB & Agg dead tests are resistant to dissolution, remaining in sediment compared with smaller forams (SM, SR, OP). This is results of dead tests, not meaning that live forams can survive in low pH environments.

**Reply:** The influence of dissolution to the foraminifera tests was mainly observed between 7.7-7.1. Before this range the physiological resistance of foraminifera was likely the most important factor as the tests were all still in good preservation conditions.

**Comment**

L387-388: only for dissolution resistant taxa (SB)

**Reply:** This information was added.

Referee#3

We thank the reviewer for helpful comments and suggestions. Bellow, you will find our answers

Referee#3 Major comments:

**Comment** - The present manuscript by Francois et al., 2022 presents a field study of Caribbean tropical foraminifera in the Puerto Morelos reef Lagoon. Six stations were sampled in October 2011 along a broad natural pH gradient generated by submarine springs. The study of spatial variability on foraminifera fauna driven by pH gradient is an original approach. This work shows interesting results that corroborate knowledge already known/suggested in previous studies. However, this manuscript deserves to be restructured and clarified on some major points before publication.

It should be kept in mind that specimens come from the natural environment (not a controlled experiment) so multiple stresses can potentially influence calcification (salinity, eutrophication, pollution, warming...) these other parameters should be further discussed according to what is known about the site in previous studies.

**Reply:** To address this comment, we now performed multiple regression analysis considering all measured variables (carbonate system, temperature, and salinity). They were compared according to their contribution to the model's Akaike Information Criterion (AIC), and the models with lowest AIC value (i.e., highest fit) were select to the analysis. Notably, consistent with previous studies we find the temperature and salinity have limited effects and the sites are not polluted so this is not a consideration. We note that the low pH and control sites are just a few meters apart hence other parameters such as light, eutrophication, pollution etc. are identical. Temperatures and salinity are only very slightly different.

We have added a discussion about salinity and for eutrophication/high nutrient concentration the following paragraph was already included in the manuscript:

"The springs from PM also have high nutrient concentrations compared to the open waters in the region (Null et al., 2014; Crook et al., 2016), however, near spring assemblages did not change towards opportunistic dominated assemblages, suggesting that the nutrient availability does not exert a major control at this site. Rather, the high-pH assemblages heavily dominated by small calcareous forms were replaced by larger symbiont-bearing species near the springs (Fig. 4a-e). Symbiont-bearing species are known to be sensitive to high nutrient loading, likely because of changes in turbidity/light regimes and their dependence on algal symbionts to enhance growth and calcification (Hallock et al., 2003). At PM despite higher nutrient levels the waters at the springs are clear and light regimes are not reduced."

Regarding warming, this is not considered to influence the species distribution as temperature changed very little and for pollution we have analyzed the concentrations of the metals (Paytan unpublished), and the concentrations were not higher at the discharge sites when compared to the control sites. We do not have a lot of heavy metal data because it is a lot of work and when we found that the concentrations at the springs were not high, we did not analyze anymore. Thus, we decided to not include this information, but we could add a sentence along the lines – preliminary data for heavy metal concentrations did not show significant differences between ojo and control sites hence we do not attribute the changes in foraminifera calcification observed to impacts of heavy metals.

**Comment** - There is a need for the bibliography to be more up to date especially those published on LBF and µCT. I suggest this non-exhaustive list: Charrieau et al., (2022); Kinoshita et al., (2021); Kuroyanagi et al., (2009); Fox et al., (2020); Iwasaki et al., (2019) …

**Reply:** We thank the referee for the suggestions and the relevant bibliography was cited in the manuscript.

**Comment** - In this paper, it is unclear about the use of live and dead fauna, if rose bengal staining has been done you must describe in the manuscript the assemblage of the foraminiferal fauna at each station and perform the ecological analyses on the live fauna. The dead fauna cannot be treated with the live fauna. If you want to study dead fauna it must be done in a separate section and clearly stated in the manuscript as "live fauna" or "dead fauna". If you study dead fauna, you must describe the assembly of dead fauna. The fact that there are few living foraminifera may be related to the seasonality of the site (previous studies?). This site may be a place of sedimentary deposition, accumulation, and currents… (1cm corresponds to what period? Previous studies?). You should discuss the sampling method used if most of the living fauna live on substrates, it could be interesting to think about a new sampling method?

**Reply:** The low number of fully stained foraminifera is probably not associated with seasonality since it is a common pattern in the Caribbean and even in pristine (off-shore) reef environments (Barbosa et al., 2009, 2012, Marine Micropaleontology) as most reef-dwelling taxa live on phytal or hard substrates rather than directly on the sediments. Most of the forams however were partially stained.

About the use of total assemblages (live + dead tests), this approach has extensively been used in the literature in the last decades to assess mid-term responses of foraminiferal assemblages to environmental conditions (e.g., nutrification and heavy metals, Barbosa et al., 2012, DOI: https://doi.org/10.2113/gsjfr.42.2.169; 2016 DOI:http://dx.doi.org/10.1016/j.marmicro.2016.07.004, Foster et al., 2012, DOI: https://doi.org/10.1016/j.marpolbul.2012.01.021), including for changes in carbonate system (e.g., Uthicke et al. 2013, Scientific reports, DOI: 10.1038/srep01769). Specifically, the present research was motivated by the results presented in Martinez et al., 2018 (Biogeosciences, DOI: 10.5194/bg-15-6819-2018), which also treated live + dead tests together to evaluate the effects of low pH/saturation state waters in the same sites analyzed here. To compare our results, focusing on mid rather than short term responses we followed the same approach.

It happens that the accumulation of foraminifera tests in the sediments integrate the effects of stressors over time (Hallock et al., 2003; Environmental Monitoring and Assessment, DOI: 10.1023/A:1021337310386), and also of small seasonal fluctuations of the assemblages, providing an ideal indicator of the foraminifera responses by reflecting the prevailing marine conditions. (Scott and Medioli, 1980; Journal of Paleoecology; DOI: http://www.jstor.org/stable/1304312). This method is supported by Stephenson et al., 2015 (Ecological indicators, DOI: http://dx.doi.org/10.1016/j.ecolind.2014.07.004), which has recently investigated the robustness of using total assemblages for ecological studies in reef environments by comparing rubble and sediment samples in Conch Reef, Florida. Buzas and Hayek, 1998 (Journal of foraminiferal research) also considered total assemblages the most adequate method for a space-time analysis. For biomonitoring studies in reef ecosystems, the use of total assemblages is the recommended to investigate foraminiferal responses over recent years or decades (Prazeres et al., 2020, and references therein, Environmental pollution, DOI: https://doi.org/10.1016/j.envpol.2019.113612).

We acknowledge that taphonomic factors such as post-mortem destruction, and transport could induce bias on our interpretations, but no significant evidence of sediment reworking and transport was found, indicating little mixing of pristine and relict tests (see the taphonomical and color analysis).

In general, the high pH stations showed no sign of ageing and little damage with "well" and "optimally" preserved tests comprising ~ 75 % of the assemblages, while the increase of dissolution and breakage was mainly associated with the springs influence. This was expected considering the overall low energy conditions of the lagoon in the protected back-reef.

About the study on phytal and rubble substrates, this would address the species responses within a particular point in time, which wasn't our objective. However, to expand our discussion we briefly discussed the stained counts, i.e., mainly dead, recently deposited tests (most had some degree of staining but were not fully brightly stained) because as stated above the species don't live in the sediments. Specifically, we mentioned the behavior of the most abundant species, *Rosalina globularis*, comparing our results with Di Bella et al. (2022) that specifically observed a resilient behavior for this genus in the low pH venting sites of Panarea.

**Comment** - You need to clarify which data are common with the paper by Martinez et al., (2018). It seems to me that you have the same dataset or a selection of them. If you share other data from this previous paper, please indicate it clearly (this can also help to reduce the manuscript).

**Reply:** The carbonate chemistry, temperature, and salinity data from 7 samples were based on those presented in Martinez et al., 2018. Here the data set was complemented with 20 mid-ranges samples collected at the same day following the same protocols described by Martinez et al. (2018) but not reported there.

As suggested, this information was added to the manuscript.

**Comment** - Your data are related to the impacts of a natural pH gradient on a series of stations at a specific date (October 2011). You are therefore looking at spatial variability of foraminifera along a pH gradient and not at temporal variability. If you want to discuss temporal projections, I will discuss this in a discussion section. To discuss temporal projections, you need to be more nuanced because you need to know the seasonal variability of the living fauna and their interannual variability and species metabolisms (maybe you have some previous studies on this site).

**Reply:** We agree with the referee that the assemblages response are associated to a spatial variability of the environmental data, but as explained above the use of total assemblages also imply a temporal (i.e., generational) factor to the assemblages response. That is the data represents the response of multiple generations of forams accumulated probably over decades (representing the upper ~ 1cm of sediment).

**Comment** - The result and discussion should be restructured, and the discussion needs major parts or titles.

**Reply:** We have restructured the manuscript and sub divided the discussion section.

**Minor comments**

**Comment** - The title does not indicate the content of the paper I would specify LBF or tropical and the study area (to be reconsidered in the light of the new orientation of the paper)

**Reply:** We have change the title to: "Acidification impacts and acclimation potential of Caribbean benthic foraminifera assemblages in naturally discharging low-pH water"

**Comment** -L41 CaCO3 = calcium carbonate

**Reply:** We changed accordantly.

**Comment** - Can you clarify what you call "small or smaller foraminifera"

**Reply:** it refers to the discussed functional groups of small miliolids and rotaliids, following the definitions of Hallock et al. (2003) for sensitivity/stress-tolerance taxa and Murray (2006) for different test compositions.

**Comment** - L209 "live fauna" or "dead fauna"? clarify this section

**Reply:** We meant total assemblages (stained + dead). It was clarified in the manuscript.

**Comment** - L211 Table S1 corresponds to Raw data of functional and test type groups. I think there is a file problem where is the faunal description?

**Reply:** We thank the referee for nothing this. The faunal description was uploaded as an excel file, since it couldn't be included in the word file due its size. It will be available for the final document.

**Comment** - L215 Considering a 3 % contribution cutoff. Why 3%?

**Reply:** For the SIMPER analysis a set of thresholds are tested and the final cutoff is defined considering the respective 2D plot with lowest stress.

**Comment** - L119 normally a minimum of 300 live foraminifera should be picked if the density is high

**Reply:** A minimum of 250 was defined following the methodology of Hallock et al. 2003, which was specifically designed for Caribbean reef ecosystems.

**Comments**

- L143 breakage and dissolution of the shells. Do you have a precise reference to do this work (quantitative approach) or is it a subjective approach?

-L154 Dissolution can affect live foraminifera and it has already been shown that some decalcified foraminifera can survive (ex. Charrieau et al 2018 Biogeosciences). To detect living foraminifera, it is either the coloration (rose bengal) or a mobility test to know if the specimen is alive or not.

**Reply:** We performed a quantitative approach using two general categories: Pristine and (2) dissolved tests, with the latter considering any degree of alteration (in most cases that was a small degree of dissolution).

**Comment** - L261 p-value = 0.00 not correct (p-value < 0.001 for example)

**Reply:** the expression was changed accordantly

**Comment** - L165 they are many papers about CTnumber please add references To compare μCT specimens, it is recommended to remove the ontogeny effect (growth-related), and therefore to compare the specimens they need to have the same size (standardized by the average of the maximum diameter of the individuals). It is always nice to see μCT on foraminifera, but you need to discuss that few individuals have been scanned and therefore be critical with the inter-individual variability.

**Reply:** The analyzed specimens presented a similar size and volume, so little or no ontogenetic effect was expected to influence the analysis and the inter-individual variability was low.

**Comment** - L248 For the CCA, one of the axes is not significant, it would be interesting to make an Ordistep preselection to select only the parameters which contribute to the CCA, and to have the two significant axes. It would be necessary to revise the design of the figure to put the variables in another color for more clarity and to put the complete legend of all the parameters used. The legends of all figures should be complete.

**Reply:** We thank the referee for this helpful suggestion**.** The second axis is probably not significant because it is related to temperature and salinity, which were the parameters that did not influence significantly the distribution of the species (BIO env/global BEST and linear regressions).

Interestingly, the Ordistep preselection indicated both variables (temperature and salinity) as still significant, and in that case, when removed from the analysis, none of the axis were significant. It probably occurred as the remaining parameters (i.e., related to the carbonate system) have high collinearity, which should be avoided for this analysis.

Since it would no longer be appropriate, we chose to remove the CCA analysis of the manuscript.

**Comments**

**-** we need to be able to understand a figure without reading the text all the time.

-L63 Fig.4e and not 4a -L284 add fig. 5 in the first sentence

**Reply:** We thank the referee for noticing these mistakes, we changed accordantly and revised the figures legends.

**List of relevant references**

- The title of the manuscript has changed to "Acidification impacts and acclimation potential of Caribbean benthic foraminifera assemblages in naturally discharging low-pH water";
- The results section was reorganized and the discussion was divided into two subsections;
- Previously published data about the lagoon circulation and sediment grain size were included in section 2.1: Study site and data retrieval;
- An explanation concerning the use of total (live + dead) assemblages was included in section 2.2: Foraminiferal analysis;
- A new taphonomical analysis was performed. Now the foraminiferal tests are classified into three the categories: 'optimally' (i.e., pristine tests), 'well' (i.e., tests with weak taphonomic signals), and 'poorly' (i.e., strongly abraded or fragmented tests) preserved, following the descriptions of Yordanova and Hohenegger, 2002;
- A color analysis was included in the manuscript to investigate the mixing of modern and relict tests.
- The analysis of similarity (SIMPER) and canonical correspondence analysis (CCA) were removed from the manuscript;
- Multiple regression analysis were included to the manuscript to investigate the relationships between carbonate chemistry and the taxonomical metrics. They were also compared according to their contribution to the model's Akaike Information Criterion (AIC).
- More information about the sampling sites (e.g., distances of each site) was included.
- A discussion about salinity and unmeasured parameters (e.g., nutrification and heavy metals) was included.
- The most important taxa for our living counts (*Rosalina globularis*) was reported in the results section and a brief discussion about our living counts concerning other natural low-pH venting sites was included.

---

## Author Response (AR2)

**Anonymous referee #3: minor revision**

After major revision, this is the new version of the manuscript by François et al. 2022. I have noted the effort made by the authors to improve the quality of the responses to the reviews, as well as the restructuring and clarification of the new manuscript. I have listed below short questions/corrections to further improve the manuscript before publication.

**Reply:** We thank the referee #3 for reviewing the document and recognizing the improvements made. Bellow, you will find the answers to the helpful comments and suggestions.

**Short questions:**

**Comment** - L 166: how did you define the surface area with a surface formula closest to the shape of the shell? perhaps add a further indication in the manuscript.

**Reply:** Foraminiferal the test area was defined following the differences in gray scales between the surface of the test (white) and background (black) in ImageJ. This information was included in l170.

**Comment** - Paragraph 3.3: it would be more logical to start the description of figure 2 with paragraph L 246-258 dealing with Fig 2a and then with paragraph L 234-245 fig 2b and c?

**Reply:** The paragraphs were reorganized accordantly.

**Comment** - Paragraph 3.4: why the shell thickness is in $mm^3$?

**Reply:** We thank the referee for noting this mistake, we changed it to mm.

**Comment** - L 161: "Discoloration patterns were analysed to investigate the vertical mixing and exposure of relict tests" do you have a data table with these observations? may be interesting to put this in SI.

**Reply:** The discoloration counts were included in table S2.

**Layout, spelling mistakes:**

**Comments** –

author affiliation, add a dot at the end

L 110: forams sampled weight, washed and dried → stained, weight, washed and dried?

L 190: was used → were used

L 218: problem of repetition in the text p=0.038

L 223: AIC=401.79 → AIC = 401.79

L 246 and others: p-value =< 0.05 → p-value ≤ 0.05

Fig. 2: some legends are shifted in the figure

L 307: \. → to remove

Fig. 4a: R2 = 0.54 (text) and R2 = 0.59 (figure) → to be homogenised

Fig 4: some legends are shifted in the figure and in the caption simbiont → symbiont

L 445: wight → weight

L 476: post mortem → italic

L 551: this species represent → represents

**Reply:** We thank the referee for noting these mistakes. We changed them accordingly.

**Comment** - L 216-218: (ρ = 0.55) → why the p is written differently?

**Reply:** Because ρ refers to the correlation coefficient and not significance.

**Comment** - L465-469: no need to repeat all μCT results make it shorter.

**Reply:** This paragraph was shortened by removing the exact values and citing fig. 6 to illustrate the differences in the μCT results.